# Regulation of CTCF loop formation during pancreatic cell differentiation

Xiaowen Lyu [1,2,3] ✉, M. Jordan Rowley [4], Michael J. Kulik [5,6], Stephen Dalton [5,6,7] & Victor G. Corces [1] ✉

Transcription reprogramming during cell differentiation involves targeting enhancers to genes responsible for establishment of cell fates. To understand the contribution of CTCF-mediated chromatin organization to cell lineage commitment, we analyzed 3D chromatin architecture during the differentiation of human embryonic stem cells into pancreatic islet organoids. We find that CTCF loops are formed and disassembled at different stages of the differentiation process by either recruitment of CTCF to new anchor sites or use of pre-existing sites not previously involved in loop formation. Recruitment of CTCF to new sites in the genome involves demethylation of H3K9me3 to H3K9me2, demethylation of DNA, recruitment of pioneer factors, and positioning of nucleosomes flanking the new CTCF sites. Existing CTCF sites not involved in loop formation become functional loop anchors via the establishment of new cohesin loading sites containing NIPBL and YY1 at sites between the new anchors. In both cases, formation of new CTCF loops leads to strengthening of enhancer promoter interactions and increased transcription of genes adjacent to loop anchors. These results suggest an important role for CTCF and cohesin in controlling gene expression during cell differentiation.

Cell differentiation requires the establishment of lineage-specific gene expression patterns[1]. Activation and silencing of transcription involve alterations in enhancer-promoter interactions that take place within, and contribute to establish, the architecture of the genome in the three-dimensional (3D) nuclear space. This architecture is created and maintained by at least two distinct processes[2–4]. One process involves the binding of transcription factors in a sequence-specific manner, which then results in the recruitment of large protein complexes that remodel or covalently modify nucleosomes and eventually bring about the recruitment of the transcription complex and/or the release of RNA polymerase II (RNAPII) into productive elongation[5]. These large complexes, present at enhancers and promoters, are composed of multivalent proteins that mediate interactions among neighboring active genes in the genome, establishing self-interacting compartmental domains[6]. Similarly, genes in a silenced state contain histone modifications such as H3K27me3 or H3K9me3, which recruit complexes such as PRC1/2 or HP1, respectively[7,8]. These complexes mediate self-interactions that establish different inactive compartmental domains. Interactions among domains in the same transcriptional state give rise to compartments, manifested in Hi-C heatmaps as the plaid pad pattern of interactions away from the diagonal and may correspond to membraneless organelles visualized by microscopy[9,10].

The second organizing process responsible for 3D chromatin architecture is the continuous cohesin extrusion taking place in the

[1]Department of Human Genetics, Emory University School of Medicine, Atlanta, GA 30322, USA. [2]State Key Laboratory of Cellular Stress Biology, Fujian Provincial Key Laboratory of Reproductive Health Research, School of Medicine, Faculty of Medicine and Life Sciences, Xiamen University, 361102 Xiamen, China. [3]Fujian Provincial Key Laboratory of Organ and Tissue Regeneration, School of Medicine, Faculty of Medicine and Life Sciences, Xiamen University, 361102 Xiamen, China. [4]Department of Genetics, Cell Biology and Anatomy, University of Nebraska Medical Center, Omaha, NE 68198, USA. [5]Department of Biochemistry and Molecular Biology, The University of Georgia, Athens, GA 30602, USA. [6]Center for Molecular Medicine, The University of Georgia, Athens, GA 30602, USA. [7]School of Biomedical Sciences, Faculty of Medicine, Chinese University of Hong Kong, Hong Kong, Hong Kong. ✉e-mail: xiaowenlyu@xmu.edu.cn; vgcorces@gmail.com

nucleus. Cohesin loads on chromatin at NIPBL sites[11,12]. During the extrusion process, cohesin may destabilize interactions between sequences in the A and B compartmental domains[13]. In doing so, cohesin brings together enhancers and promoters located within the same extrusion loop or adjacent to loop anchors. Cohesin extrusion stops at CTCF sites when arranged in a convergent orientation, creating loops anchored by CTCF[14–20]. These loops can form insulated neighborhoods that preclude interactions between regulatory sequences located inside and outside of the loops[21]. CTCF loops can also stably tether enhancers to their cognate promoters[22–24] and contribute to the establishment of specific fates during cell lineage commitment[25–28].

Therefore, the establishment and disassembly of enhancer-promoter interactions required to create patterns of gene expression necessary for cell fate transitions must involve changes in the biochemical and biophysical forces that create compartmental domains and their interactions with other compartmental domains as well as in the extrusion process mediated by cohesin and the pausing of this process by CTCF. The establishment of different cell fates during development involves the activation of signaling pathways that activate transcription factors, leading to changes in transcription and chromatin-associated proteins. This in turn will alter compartmental domains and the interactions that create compartments. Less clear is how cells control the recruitment of CTCF to new sites in the genome and the formation of new loops anchored by CTCF and/or other proteins[29]. Most CTCF sites are present at the same sequences in a variety of cultured cell lines[30] but approximately 25% are cell-type specific and change during the establishment of various cell lineages[25,31–37]. Interestingly, a subset of variable CTCF sites are present at transposable elements and contribute to differences in gene expression among cell types[38]. CTCF loops are present simultaneously in a large fraction of cells in a population as suggested by the intensity of the corner dots present in Hi-C heatmaps that allow their identification[39]. Since CTCF loops contribute to the establishment and regulation of enhancer-promoter interactions, their formation or dissolution must be closely coordinated with the activation or dismissal of the regulatory regions they control. To explore the contribution of these two principles of 3D genome organization, interactions among compartmental domains and cohesin extrusion, to the regulation of changes in transcription during cell fate transitions, we analyzed alterations in nuclear architecture taking place during the differentiation of human embryonic stem cells (hESCs) into pancreatic cells[40]. Following the specification of the definitive endoderm (DE) after gastrulation, the primitive gut tube (PGT) forms by migration and involution. Pancreatic progenitors (PP) are then committed in a fraction of PGT cells, within which islet progenitors of five major endocrine cell types, including insulin-secreting β cells, are determined before the maturation of prenatal endocrine cells[41]. Mechanistic studies of embryonic pancreatic β cell specification and maturation have been hampered by the low cell numbers available from embryo dissection. Thus, in vitro differentiation protocols to derive functional pancreatic islet cluster cells from embryonic stem cells have been used as a substitute to chart cellular identities and to dissect molecular mechanisms of embryonic pancreas development[42]. Using culture systems, it has been shown that pancreas-specific enhancers undergo resolution of poised states by stepwise loss of H3K27me3 and gain of H3K4me3 during human embryonic stem cell differentiation into endocrine lineages[43]. In addition, H3K9me3 has been reported to be transiently deposited in intermediate endoderm stages and erased in the process of final commitment to pancreatic lineages[44], and mutation of the transcriptional repressor REST results in an increase of pancreatic endocrine cells[45]. ATAC-seq and single-cell RNA-seq analyses have shown that key lineage-defining loci are epigenetically primed before activation and widespread epigenome remodeling occurs during differentiation[46–48]. Results from these studies suggest a complex interplay between enhancers, repressors, and chromatin modifications in the differentiation of pancreatic cells. Furthermore, analyses of pancreatic islet enhancers affected by SNPs associated with type 2 diabetes and their target genes have provided critical insights into chromatin regulatory processes involved in pancreatic cell fate[47,49]. Despite these important advances, we currently lack a mechanistic understanding of how these changes in enhancer function and histone modifications take place during pancreatic cell differentiation in the context of the 3D organization of the chromatin and how CTCF and cohesin extrusion control enhancer-promoter targeting to elicit different patterns of gene expression.

Here we utilize in vitro pancreatic cell differentiation to mimic human pancreatic endocrine lineage commitment and we analyze changes in compartmental interactions and CTCF loops during the differentiation process. The results provide insights into how 3D genome architecture interplays with cohesin extrusion and the formation or disassembly of CTCF loops to regulate enhancer-promoter interactions required for the differentiation of pancreatic cells.

## Results
### Redistribution of CTCF loops during pancreatic cell differentiation
To understand the contribution of CTCF-based 3D chromatin organization to the establishment of cell fate during development, we induced the differentiation of H9 hESCs into pancreatic islet organoids and isolated several cell fate transition intermediates, including definitive endoderm (DE), primitive gut tube-like (PGT), pancreatic progenitors (PP), and stem cell-derived β-cell organoids (SC-β organoids) following established protocols[42,48,50]. Quality controls showing the presence of specific cell types based on the expression of known gene markers at different steps of the differentiation process are shown in Supplementary Fig. 1. To further analyze the reproducibility of the differentiation process, we performed RNA-seq in two independent replicates for each differentiation stage. We found that independent differentiation replicates show high correlation of gene expression based on qPCR of RNA (Supplementary Fig. 2a), each differentiation stage expresses the expected marker genes (Supplementary Fig. 2b), and the expression of marker genes is highly correlated between replicates (Supplementary Fig. 2c). To analyze changes in 3D organization during differentiation of hESCs into pancreatic cells, we performed in situ Hi-C[51] and obtained between 700 and 1000 million contacts from two replicates for each stage after quality filtering. Supplementary Data 1 contains information on the different quality control steps of Hi-C data processing for all samples. Additional information showing reproducibility between replicates for each differentiation stage is shown in Supplementary Fig. 2d, e.

We then analyzed the Hi-C data to identify changes in CTCF loops, defined as point-to-point interactions visible as punctate signal (corner dots) in Hi-C heatmaps, at 5 kb and 10 kb resolution using SIP[52]. Loops visualized in Hi-C heatmaps as punctate signal are generally caused by stopping of cohesin extrusion at CTCF sites arranged in a convergent orientation[18,51]. Although we will refer to these loops as "CTCF loops" throughout the manuscript, not all of them contain CTCF at one or both anchors and a subset could be formed by obstruction of cohesin extrusion by other proteins[52,53]. Loops detected based on the presence of corner dots are not necessarily the same as Topologically Associating Domains (TADs), which are identified using algorithms that detect changes in the directionality of interactions and do not always have CTCF sites at their boundaries[54]. We identified a total of 40,633 loops from all differentiation stages combined. Of these, 29,905 loops persist unchanged throughout differentiation whereas the remaining 10,728 are altered during the transitions between consecutive stages. For example, a subset of loops is lost or gained when H9 hESCs differentiate into DE, a different group of loops is altered when DE cells differentiate into PGT, etc. Using meta-analysis of interaction scores

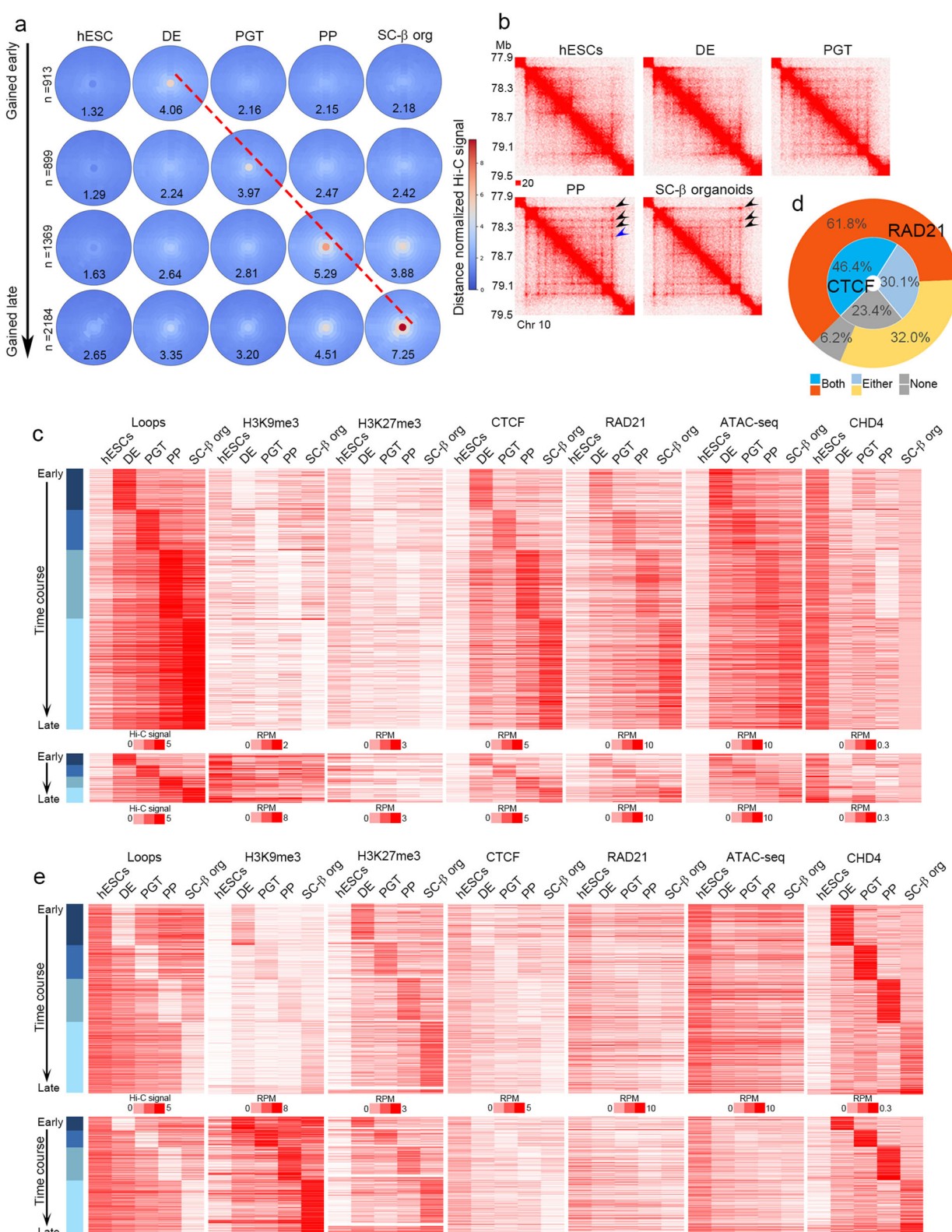

for loops that behave dynamically during pancreatic differentiation, we identified stage-specific loops that are not present at one stage, are formed in the transition to the following stage, and are then disassembled when the cells differentiate further (Fig. 1a). The number of stage-specific loops is approximately the same for each stage, except for the SC-β organoid stage in which twice as many loops are altered. Of the 10,728 altered loops, 5365 are gained (Fig. 1a) and 5363 are lost

(Supplementary Fig. 3a) during differentiation. Loops gained in pancreatic progenitor and islet cells have much higher changes in contact frequencies calculated by aggregate peak analysis (APA) scores of distance-normalized Hi-C interactions than those gained at earlier stages (Fig. 1a). APA histograms showing the distribution of fold changes of loop signals for all loop classes are shown in Supplementary Fig. 3b. These changes in APA scores are also observed when using Hi-C

**Fig. 1 | Analysis of stage-specific CTCF loops during the differentiation of hESCs into pancreatic cells. a** Aggregate peak analysis (APA) of Hi-C data obtained in cells at different stages during the differentiation of H9 hESCs into SC-β organoids. Each circle represents the aggregate of the signal present in all corner dots corresponding to CTCF loops detected by SIP at each stage (see Methods). Each row shows CTCF loop APA values in cells at different stages for CTCF loops specific for each stage. For example, the top row shows APA values of DE-specific CTCF loops as well as the APA values of these CTCF loops at other stages of differentiation. The results suggest that cells at each differentiation stage form new CTCF loops that are not present in previous or subsequent stages. **b** Specific example of a region of chromosome 10 showing CTCF loops absent in hESCs, DE, and PGT; formed in PP and maintained in SC-β organoids (black arrowheads); or formed in PP but lost in SC-β organoids (blue arrowhead). **c** Heatmaps showing formation of stage specific CTCF loops. Loops have been divided into two groups, those containing low levels of H3K9me3 at the previous stage (top) and those containing high levels of

H3K9me3 at the previous stage (bottom). These results show that 253 stage-specific CTCF loops contain H3K9me3 whereas 5033 lack this histone modification. The formation of CTCF loops at each specific stage correlates with increased levels of CTCF, although a subset of loops already contains CTCF before loop formation. Establishment of new loops correlates with increased RAD21 and ATAC-seq signal and decrease of CHD4, H3K9me3, and H3K27me3. **d** Analysis of the distribution of CTCF and RAD21 at CTCF loops. The diagram describes the fraction of loops containing CTCF or RAD21 at one, both or no loop anchors. **e** Heatmaps showing disassembly of stage specific CTCF loops. Loops are divided into two groups, those containing low (top) or high (bottom) levels of H3K9me3 at the stage when they are disassembled. Dismantling of CTCF loops at each specific stage correlates with decreased levels of CTCF, RAD21, and ATAC-seq signal, and increase of H3K9me3, H3K27me3, and CHD4. Abbreviations: human embryonic stem cells (hESCs), definitive endoderm (DE), primitive gut tube-like (PGT), pancreatic progenitors (PP), and stem cell-derived β-cell organoids (SC-β organoids).

interactions that are not distance-normalized (Supplementary Fig. 3c). An example of a series of nested loops that increase in strength during pancreatic cell differentiation at the PP and SC-β organoid stages is shown in Fig. 1b. In addition to loops that are made at specific stages, different sets of CTCF loops present at each stage are disassembled in the transition to subsequent stages as differentiation proceeds (Supplementary Fig. 3a). In general, loops that decrease in interaction frequency during differentiation do so only slightly, and the change in interaction frequency is not as pronounced as for those that gain strength (Supplementary Fig. 3a).

One possible explanation for the formation and disassembly of CTCF loops during cell differentiation is changes in the occupancy of CTCF at specific sites in the genome. Several mechanisms have been suggested to regulate CTCF binding to DNA, including covalent modifications of CTCF, DNA methylation and the recruitment of the ChAHP complex[37,55,29,56,57]. This complex, which is composed of CHD4, ADNP and the heterochromatin protein HP1, has been shown to compete with CTCF for a subset of genomic sites[56]. To examine the relationship between changes in CTCF loops and changes in the occupancy of CTCF, other transcription factors (TFs), or the ChAHP complex, we performed ATAC-seq as well as ChIP-seq with antibodies to CTCF, RAD21, H3K9me3, and H3K27me3, and CUT&Tag with antibodies to CHD4. CTCF anchors for this analysis were defined as 10 kb sequences surrounding the region containing corner dots in Hi-C data. After initial analysis of the data, we noticed that anchors of CTCF loops that form at specific stages of pancreatic cell differentiation are located in regions containing variable levels of H3K9me3 at the previous differentiation stage. Therefore, we separated dynamic CTCF loops into two categories, those containing low levels of H3K9me3 at the previous stage (Fig. 1c, top) and those containing high levels of this modification (Fig. 1c, bottom). These results show that 253 stage-specific CTCF loops contain H3K9me3 whereas 5033 lack this histone modification. An example showing quantitative differences in the levels of H3K9me3 between these two types of anchors is shown in Supplementary Fig. 3d. For both classes of loops, increased interactions between loop anchors at each differentiation stage correlates with loss of H3K9me3 (Fig. 1c). The loss of H3K9me3 takes place in proximity to CTCF sites but not at enhancers or TSSs of adjacent genes or at random regions of the genome (Supplementary Fig. 3e). Interestingly, the same correlation can be observed with loss of H3K27me3, which we discuss in more detail below. Approximately 77% of new stage-specific loops contain CTCF at one or both anchors whereas 94% contain RAD21 (Fig. 1d). Therefore, formation of new CTCF loops during pancreatic cell differentiation may be due, at least in part, to changes in CTCF and/or RAD21 occupancy, a result that is confirmed by the recruitment of these two proteins to stage-specific anchors and their dismissal when the loops disassemble in the following stage (Fig. 1c). The presence of CTCF, RAD21, and perhaps other transcription factors, at loop anchors is associated with chromatin accessibility

measured by ATAC-seq (Fig. 1c). Recruitment of CTCF to stage-specific loops also correlates with the loss of CHD4 at the same stage when CTCF and RAD1 are gained (Fig. 1c). The opposite correlations are observed at loops that disassemble during the transition between specific cell differentiation stages (Supplementary Fig. 3a). These loops can be also divided in two classes, those whose elimination correlates with the presence of low levels of H3K9me3 and H3K27me3 at loop anchors (Fig. 1e, top) and those at which levels of H3K9me3 and H3K27me3 are high (Fig. 1e, bottom). Dismissal of CTCF loops at each differentiation stage correlates with a decrease in the levels of CTCF and RAD21 and increased recruitment of CHD4 (Fig. 1e). Loops that dissolve at specific differentiation stages are present in regions devoid of H3K9me3 and H3K27me3 at the previous stage, but these loop anchors gain one or both of these two modifications concomitant with their removal (Fig. 1e). These observations suggest an inverse correlation between the recruitment of CTCF/cohesin to loop anchors and the presence of H3K9me3, H3K27me3, and the ChAHP complex at different stages of pancreatic cell differentiation.

## Establishment of CTCF loops in sequences containing H3K9me3 correlates with loss of compartmental interactions

Regions of the genome containing H3K9me3 interact with each other to form biomolecular condensates. Although the exact mechanism is not understood, these condensates may form through liquid-liquid phase separation that may be mediated by intrinsically disordered regions in HP1a[58–60]. Active regions of the genome also interact, although the formation of these interaction hubs may not involve phase separation[61]. These interactions can be visualized in Hi-C heatmaps by the checkerboard signal away from the diagonal corresponding to interactions among compartmental domains containing H3K9me3[10]. Changes in H3K9me3 during pancreatic cell differentiation that accompany activation of new CTCF loop anchors should thus result in alteration of these interactions, which should be visible in Hi-C heatmaps as compartmental changes. Therefore, the observation of an inverse relationship between changes in CTCF loops and changes in H3K9me3 suggests that the formation of new CTCF loops may be accompanied by loss of compartmental interactions and vice versa. To explore this possibility, we first called A/B compartments using Principal Component Analysis (PCA) and 25 kb bins to search for switches in compartmentalization at high resolution. This resolution results in the identification of smaller genomic intervals belonging to A and B compartments compared to the standard 1 Mb resolution[62], allowing the identification of compartmental changes during pancreatic cell differentiation, both by changes between A and B and by changes in the magnitude of the Eigenvector positive or negative values. Genomic intervals belonging to the A compartment, forming A compartmental domains, are defined as regions with positive Eigenvector values and correlate with the presence of ChIP-seq signal for RNA Polymerase II phosphorylated in serine 2 (RNAPIISer2ph) or the histone

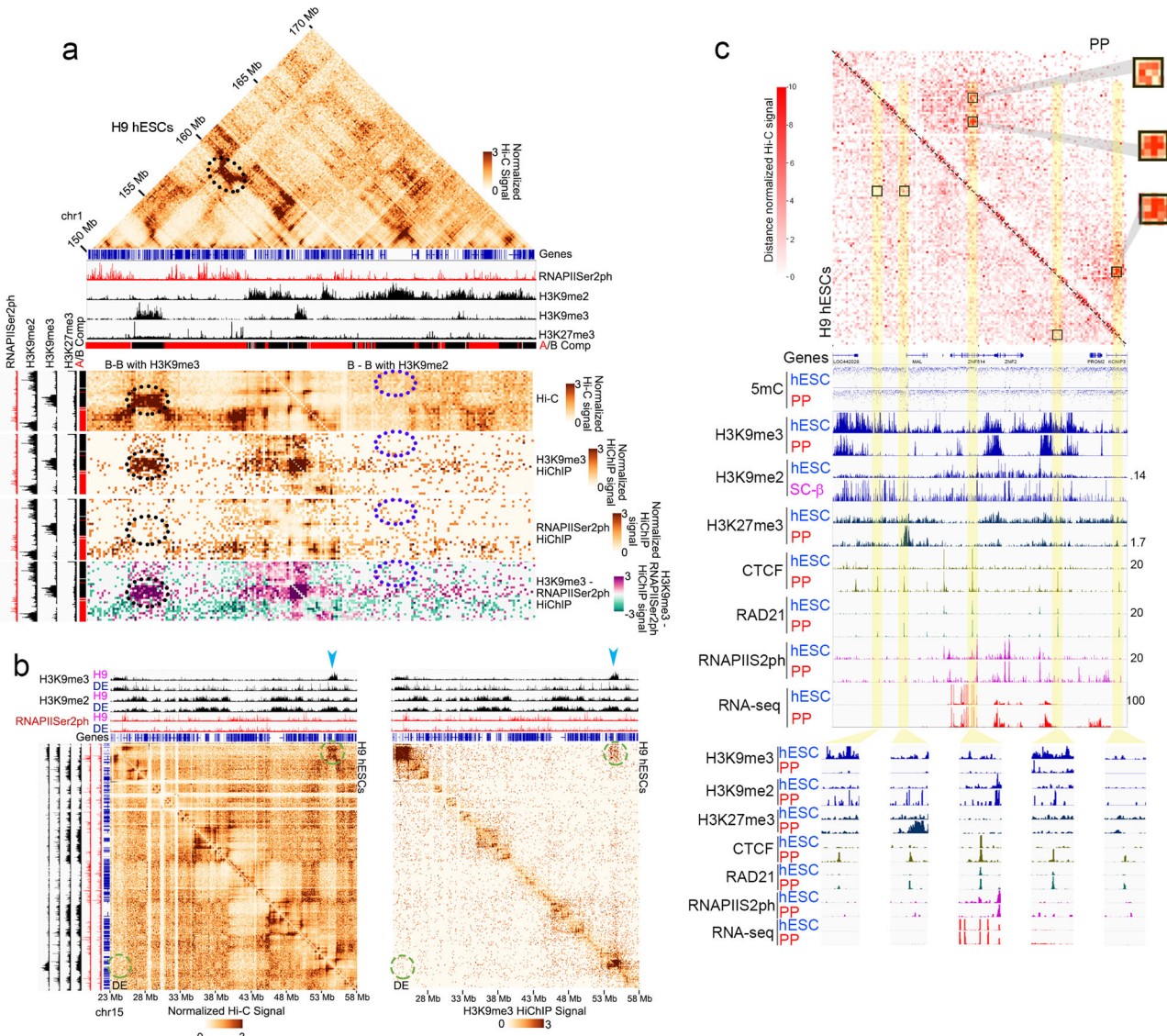

**Fig. 2 | Correlations between changes in CTCF loops and compartmental interactions. a** Compartmental interactions in a 20 Mb region of chromosome 1 in H9 hESCs. The top of the figure shows a Hi-C heatmap, the correlation between compartmental interaction signals and A/B compartment calls, and the presence of RNAPIISer2ph, H3K9me3, H3K9me2, and H3K27me3. Regions containing high levels of H3K9me3 (dotted black ellipse) or high levels of RNAPIISer2ph interact frequently whereas those containing H3K9me2 show low interaction frequencies. The bottom part of the figure shows a subset of interactions in the same region of the genome. Very frequent interactions mediated by H3K9me3 (dotted black ellipse) and rare interactions between regions containing H3K9me2 both map to the B compartment. The results are confirmed by H3K9me3 and RNAPIISer2ph HiChIP. **b** Compartmental interactions observed in Hi-C heatmaps (green circles) between regions containing H3K9me3 in H9 hESCs (blue arrowhead) are lost in DE

cells when H3K9me3 is replaced for H3K9me2 in these cells. HiChIP experiments with H3K9me3 antibodies shown in the right panel confirm these results.
**c** Example of changes in CTCF loops observed by Hi-C between H9 hESCs and PP cells in a region of chromosome 2. Three loops (black squares) present in PP cells are not observed in H9 hESCs. The two loops on the left form between two new PP CTCF sites and one CTCF site that is also present in H9 cells. The loop on the right forms between two new CTCF sites present in PP but absent in H9 cells. Formation of the loops correlates with the presence of RAD21 and changes in transcription of surrounding genes. Loop formation also correlates with decreased H3K9me3 and increased H3K9me2 at loop anchors. No changes in DNA methylation are observed at this resolution. Abbreviations: human embryonic stem cells (hESCs), definitive endoderm (DE), primitive gut tube-like (PGT), pancreatic progenitors (PP), and stem cell-derived β-cell organoids (SC-β organoids).

modifications H3K4me3 and H3K27ac (Fig. 2a)[10]. B compartmental domains correlate with the presence of silenced genes or absence of genes. Depending on the chromosome, B compartmental domains may contain H3K9me3, H3K9me2, or H3K27me3, and the strength of interactions between B compartmental domains depends on which one of these histone modifications is present in the interacting sequences (Fig. 2a). Interactions among B compartmental domains are stronger in regions containing H3K9me3 (Fig. 2a). To confirm these observations we performed HiChIP analyses with antibodies against H3K9me3 or RNAPIISer2ph. The results show that regions containing

H3K9me3 interact strongly, as also observed in Hi-C data, whereas B compartmental domains lacking H3K9me3 but containing H3K9me2 interact less frequently (Fig. 2a, bottom panels).

Compartmental interactions between H3K9me3-containing regions detected by Hi-C or by H3K9me3 HiChIP change dynamically during cell differentiation (Fig. 2b). For example, when H3K9me3 is lost in the transition from H9 hESCs to DE, H3K9me2 is gained in the same region, and compartmental interactions decrease in frequency (Fig. 2b). A second example is shown in Supplementary Fig. 4a, where the loss of an H3K9me3 domain concomitant with the gain of

H3K9me2 in PP cells results in a decrease of compartmental interactions compared to hESCs (blue boxes in Supplementary Fig. 4a). Subtraction heatmaps between the Hi-C heatmaps obtained in PP and H9 hESCs further highlight these changes (Supplementary Fig. 4b). Interestingly, the changes are not accompanied by large changes in DNA methylation over broad domains of chromatin containing H3K9me3 (Supplementary Fig. 4a) but could be related to fine-scale alterations in DNA methylation (see below). These regions also gain CTCF and cohesin, which in turn form loops via cohesin extrusion to establish point-to point interactions. These new CTCF sites are located in the region of Supplementary Fig. 4b highlighted by a gray bar, which is enlarged in Fig. 2c for easier visualization. Hi-C heatmaps show the presence of two CTCF loops in PP cells on the left side of the map and a third one on the right (Fig. 2c, black squares). The two loops on the left form between two new CTCF sites oriented towards the right that appear in PP cells in regions that lose H3K9m3 and/or H3K27me3 with a pre-existing CTCF site oriented towards the left. RAD21 is present at the anchors of these loops in PP cells, presumably because its extrusion is stopped by the presence of CTCF (Fig. 2c). The loop on the right side of the heatmap is formed between two CTCF sites in convergent orientation that are not present in hESCs and appear de novo in PP cells. The formation of this loop is accompanied by an increase in RNAPIISer2ph and activation of expression of the *PROM2* gene (Fig. 2c). Changes in interactions leading to the formation of new CTCF loops during pancreatic cell differentiation can be better appreciated in subtraction heatmaps of the two cell stages (Supplementary Fig. 4c). In conclusion, replacement of H3K9me3 by H3K9me2 during pancreatic cell differentiation correlates with loss of compartmental interactions. It is thus possible that the simple switch from H3K9m3 to H3K9me2 may lead to the escape of the affected sequences from H3K9me3 biomolecular condensates. At the same time, CTCF is recruited to sites present in these sequences to form point-to-point interactions via cohesin extrusion, which accompany activation of transcription.

## New CTCF loops can be established by different mechanisms
Formation of new loops at different stages of pancreatic cell differentiation appears to broadly correlate with increased occupancy by CTCF (Fig. 1c). However, in principle, new loops established in a specific stage could form between pre-existing CTCF sites, between new stage-specific sites, or between both types. Visual inspection of CTCF ChIP-seq data suggests that some CTCF sites remain constant during differentiation into pancreatic lineages whereas others are dynamic and appear or disappear at different stages (Fig. 3a). To explore this issue in detail, we analyzed changes in loop anchors for all stage specific loops in the context of changes in CTCF occupancy. We find that only 8-17% of stage-specific loops are formed between two new CTCF sites that were not present in the previous differentiation stage. The rest of the new loops are formed between sites in which at least one of the anchors contains a pre-existing CTCF site not previously involved in the formation of a loop (Fig. 3b). Some anchors forming loops in one differentiation stage also form loops in the following stage that are arranged differently with respect to the previous loops (Fig. 3b). Depending on the stage, around 60% percent of new loops maintain one of the CTCF anchors but now this site forms a loop with a CTCF site further distal than the previous one (Fig. 3b). A second group of 20–30% of new loops form between pre-existing anchors that pair with new anchors to form two shorter or two longer loops (Fig. 3b). These results suggest that most stage-specific loops are formed by increasing in size, as if cohesin released from the original anchor is stopped at new CTCF sites present in later stages (Fig. 3c). Representative examples of new loops formed by this logic during pancreatic cell differentiation are shown in Fig. 3d, e. In the first example, a CTCF loop present in H9 hESCs is maintained in SC-β organoids (Fig. 3d, blue arrowhead). This loop appears to

extend past one of the original anchors and forms two loops with two new anchors in SC-β organoids (black arrowheads) but not in H9 hESCs (black circles). One of the new anchors involves de novo recruitment of CTCF in SC-β organoids whereas the second anchor contains CTCF in both differentiation stages but does not form a loop in H9 hESCs (Fig. 3d). A second example involves strengthening of a loop by de novo recruitment of CTCF in SC-β organoids to one of the anchors of a weak loop present in H9 hESCs (Fig. 3e, blue arrowheads). This anchor forms larger loops with existing CTCF sites in SC-β organoids that were not previously forming loops in hESCs (Fig. 3e, black arrowheads). These new anchors are bound by CTCF in both SC-β organoids and hESCs based on ChIP-seq results but have dramatically increased ATAC-seq signal in SC-β organoid cells. This increased accessibility is suggestive of recruitment of other proteins to these anchors.

## Formation of new loops correlates with stage-specific recruitment of transcription factors to loop anchors
Formation and disassembly of CTCF loops correlates with changes in H3K9me3 and H3K27me3 in loop anchors defined as 10 kb regions (Fig. 1c, e). To gain additional insights into the mechanisms by which CTCF loop anchors change during pancreatic cell differentiation, we determined changes in these two histone modifications as well as DNA methylation at the actual CTCF sites present at these loop anchors. The results show that, for each stage in which a new loop is established, H3K9me3 decreases in the immediate region where the CTCF site is located (Fig. 4a). The same is the case for H3K27me3, although the differences in the levels of this modification between consecutive differentiation stages at CTCF sites involved in making new loops is not as pronounced as for H3K9me3. At this level of resolution, there are also dramatic changes in DNA methylation in a 600 bp region surrounding the CTCF sites, with a pronounced drop in 5mC that correlates with the decrease in the two repressive histone modifications (Fig. 4a). These changes in DNA methylation may regulate CTCF recruitment to these dynamic sites, since the interaction of CTCF with DNA is sensitive to methylation at genomic sites containing CpG at position 2 of its binding motif[57,63,64]. The results suggest a strong correlation between binding of CTCF at stage specific anchors and hypomethylation of these anchors. The same sites are hypermethylated in previous and subsequent stages in which CTCF levels at these anchors decrease (Fig. 4a). It is worth emphasizing that these changes in DNA methylation affect a small region surrounding the CTCF binding site, rather than large chromatin domains as noted above (Fig. 2c). To explore the roles of H3K9me3 versus H3K27me3 in more detail, we separated CTCF loop anchors into those that overlap with H3K9me3 peaks called by MACS and those that overlap with H3K27me3 peaks. CTCF loop anchors overlapping each histone modification also contain smaller amounts of the second one but the overall pattern of changes and overlap with changes in DNA methylation during pancreatic cell differentiation is similar to that found for the combined CTCF anchors (Supplementary Fig. 5a, b).

We then explored the possibility that other proteins may be responsible for eliciting these changes to allow binding of CTCF. We used ATAC-seq obtained from cells in the various differentiation stages and separated paired reads into those containing fragments in the 50-115 bp range (ATAC-TF), which correspond to bound transcription factors, from those in the 180-247 bp range, which map the location of nucleosomes (ATAC-Nuc). We then examined the distribution of ATAC-seq signal at CTCF sites that change at different stages of pancreatic cell differentiation and are present at dynamic loop anchors (Fig. 4a). In general, levels of CTCF at stage-specific anchors are lower at previous stages and decrease again at subsequent stages, and the ATAC-TF signal changes in parallel (Fig. 4a). These sites are flanked by well-positioned nucleosomes based on ATAC-Nuc signal in DE, PGT, and PP cells, and the nucleosomes remain positioned when CTCF

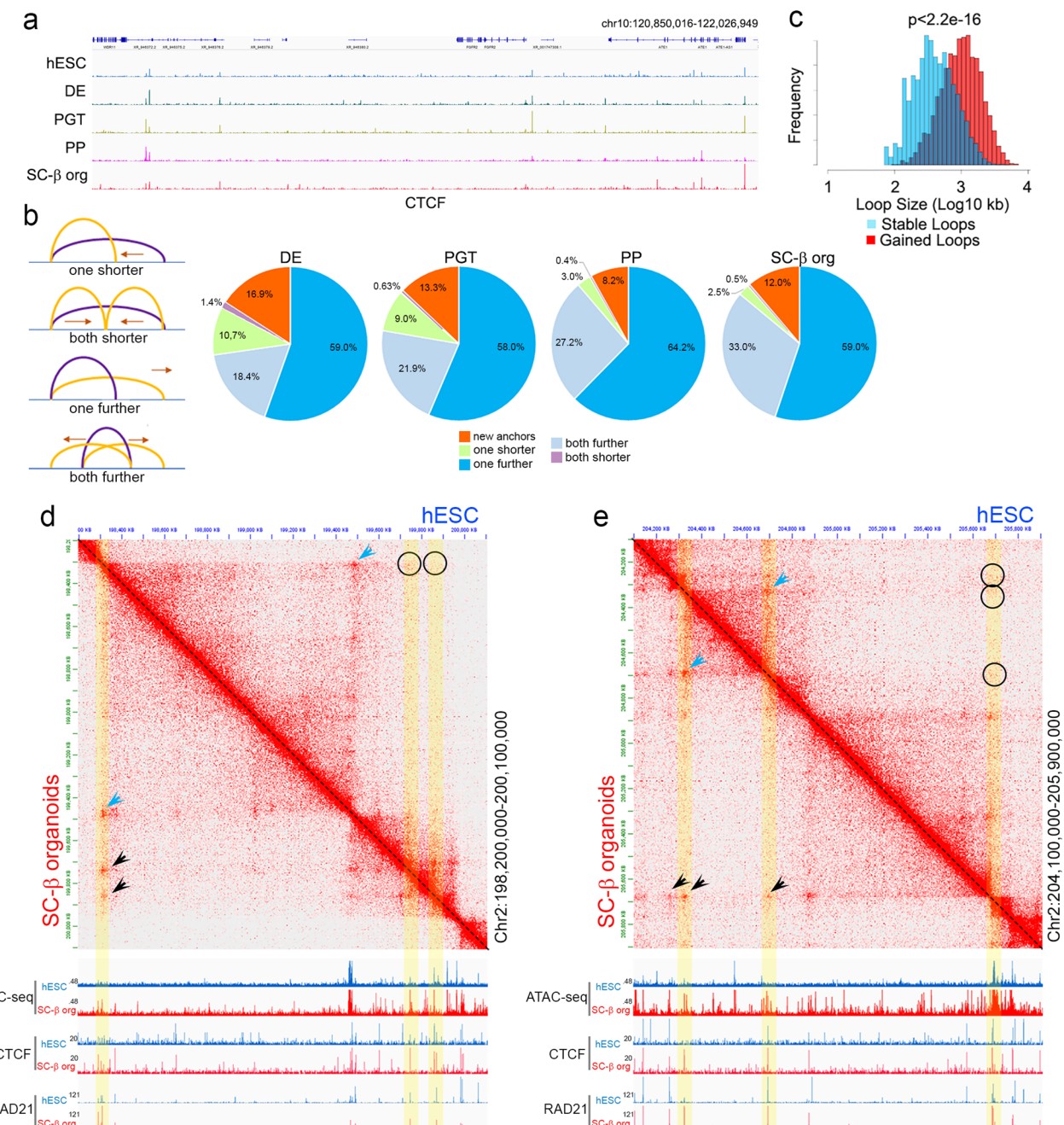

**Fig. 3 | Changes in CTCF loops at different stages during the differentiation of H9 hESCs into pancreatic cells. a** Example showing changes in the distribution of CTCF in a region of chromosome 10 at different stages of differentiation. **b** Different processes by which CTCF loops change during differentiation by moving the location of one or both loop anchors. **c** Comparison of the number and size of stable versus new loops formed during pancreatic cell differentiation. **d** Example of CTCF loops present in SC-β organoids but not in H9 hESCs. New loops (black arrowheads) form by extension of one of the anchors of an existing loop (blue arrowhead). The new loop anchors in SC-β organoids also contain CTCF in H9 cells but fail to form loops. **e** A second example of CTCF loops present in SC-β organoids but not in H9 hESCs. New loops (black arrowheads) form by extension of one of the anchors of an existing loop (blue arrowheads). One of the new loops forms by recruitment of CTCF to a new genomic site whereas two other loop anchors already contain CTCF in H9 cells but fail to form loops. These anchors contain increased ATAC-seq signal when they form loops. Abbreviations: human embryonic stem cells (hESCs), definitive endoderm (DE), primitive gut tube-like (PGT), pancreatic progenitors (PP), and stem cell-derived β-cell organoids (SC-β organoids).

levels and ATAC-TF signal decreases at later stages. Surprisingly, flanking nucleosomes are less positioned in SC-β organoids (Fig. 4a).

The ATAC-TF signal follows a similar pattern to that of CTCF in dynamic loop anchors during pancreatic cell differentiation. To explore the possibility that other proteins are recruited to CTCF loop anchors to elicit changes in DNA methylation and/or histone modifications, we examined the presence of specific transcription factor binding motifs at the summits of ATAC-TF peaks at stage-specific

anchors. We find that CTCF is the most significantly enriched motif on anchors of loops gained at each specific stage. In addition, pioneer TFs such as FOXA2, and FOXA1 are significantly enriched at anchors of loops gained in DE cells, HNF1 and GATA6 in PGT, and FOXA1, HNF1, and RFX4 in SC-β organoids (Fig. 4b). These transcription factors have been shown to play a role in the differentiation of endodermal derivatives[65–67]. Binding motifs for FOXA2 are enriched in the region surrounding CTCF motifs at stage-specific anchors, supporting the

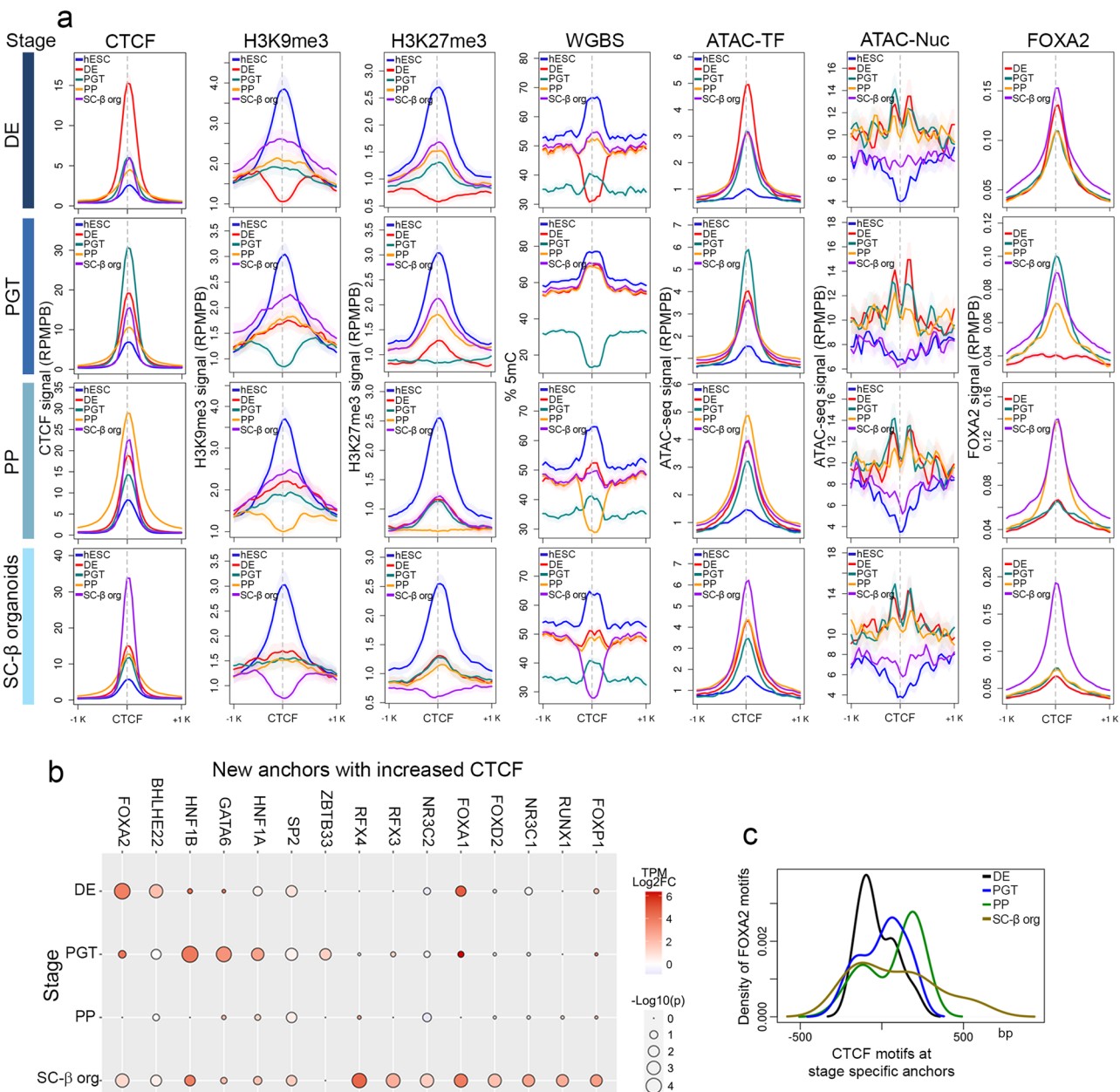

**Fig. 4 | Changes in chromatin accessibility at new CTCF loop anchors during cell differentiation. a** Stage-specific loops in which the loops form by de novo recruitment of CTCF to one or both anchors. Each row represents one differentiation stage. The first column shows the levels of CTCF from ChIP-seq experiments at anchors of loops specific for each stage. Levels of CTCF are highest at the stage when the loop is detected by Hi-C. The ATAC-TF signal, corresponding to subnucleosomal reads in the 50–115 bp, represents bound TFs and varies following a similar pattern to CTCF. CTCF sites are flanked by positioned nucleosomes (ATAC-Nuc signal corresponding to ATAC-seq reads 180–247 bp long) at most stages and the level of DNA methylation decreases at each stage in a small region

surrounding the CTCF site concomitant with the presence of this protein. Levels of FOXA2 increase at stage specific loop anchors for all stages except SC-β organoids; loop anchors for this stage contain FOXA2 starting at DE **b** Frequency of binding motifs for various transcription factors found at the summits of ATAC-TF peaks present within 10 kb regions containing anchors of CTCF loops showing increased interactions at different stages of pancreatic cell differentiation. **c** Distribution of FOXA2 motifs with respect to CTCF motifs present at stage specific loop anchors. Abbreviations: human embryonic stem cells (hESCs), definitive endoderm (DE), primitive gut tube-like (PGT), pancreatic progenitors (PP), and stem cell-derived β-cell organoids (SC-β organoids).

idea that this and other pioneer factors may be required for increased recruitment of CTCF to these loop anchors (Fig. 4c). To further test the actual presence of these transcription factors at CTCF loop anchors, we performed ChIP-seq in DE, PGT, PP, and SC-β organoid cells with antibodies to FOXA2. Results suggest that FOXA2 is present in DE cells at DE-specific loop anchors and its levels decrease at these anchors in PGT cells when these DE anchors are not involved in loop formation (Fig. 4a). A similar result can be observed in other differentiation

stages, with FOXA2 levels at CTCF loop anchors increasing at each specific stage. An exception is SC-β organoid specific loop anchors, which also contain high levels of FOXA2 at all previous stages when CTCF protein is not bound, and the anchors are not forming loops (Fig. 4a). These results suggest that FOXA2, in addition to facilitating the recruitment of lineage-specific transcription factors during the differentiation of endodermal tissues, may also facilitate CTCF recruitment. These correlative observations suggest that recruitment

of CTCF to new sites in the genome may first require the recruitment of pioneer factors, which position nucleosomes in the flanking regions and contribute to local DNA demethylation, thus allowing binding of CTCF. FOXA2 has been shown to interact with TET1 to induce local DNA demethylation[68,69]. However, the observation that this protein is already bound to SC-β organoid specific loop anchors at early stages of differentiation that remain methylated until the SC-β organoid stage suggests that additional factors may be required to trigger demethylation of these anchors.

A subset of new loop anchors does not appear to be bound by CTCF based on the absence of peaks in CTCF ChIP-seq experiments (Fig. 1d). To explore the possibility that other TFs may interfere with cohesin extrusion at these sites, we pooled all ATAC-seq peaks present in 10 kb loop anchors lacking CTCF and examined the presence of TF binding motifs at the summits of ATAC-TF peaks. The results suggest enrichment of motifs for pioneer factors such as FOXA2 (Supplementary Fig. 6a) but also stage-specific TFs that, perhaps in conjunction with RNAPII, could interfere with cohesin extrusion as recently demonstrated[53,70] (Supplementary Figure 6b).

## Targeting enhancers to lineage-specific genes within regions of extended CTCF loops

Differentiation of cell types during development requires the activation of new enhancers and the establishment of new enhancer-promoter interactions. To gain further insights into the relationship between changes in CTCF loops and gene expression during pancreatic cell differentiation, we performed HiChIP for RNAPIIS2ph, H3K4me1, and H3K27ac in H9 hESCs, DE, PGT, PP, and SC-β organoid cells. We first used self-ligation events to determine the distribution of these histone modifications and RNAPIIS2ph in the genome[71]. We then identified active enhancers in the regions surrounding CTCF loop anchors at each differentiation stage based on the presence of ATAC-seq, H3K4me1, and H3K27ac peaks. Active promoters were identified by the presence of H3K27ac at mapped TSSs. New enhancers activated at each stage are located in regions of accessible chromatin as indicated by ATAC-TF signal mapping the presence of transcription factors. These regions also contain H3K4me1, H3K27ac, and RNAPIIS2ph (Fig. 5a and Supplementary Fig. 7a). Stage-specific enhancers lack these chromatin characteristics at the previous stage of pancreatic cell differentiation, and they lose them again at the following stage. For example, enhancers active in DE cells are bound by transcription factors (ATAC-TF signal) and contain H3K4me1 and H3K27ac in DE but not in hESCs (Fig. 5a and Supplementary Fig. 7a). The opposite is the case for both H3K27me3 and H3K9me3. A similar pattern is observed for promoters of genes activated in a stage-specific manner (Fig. 5a and Supplementary Fig. 7a). The opposite is observed for enhancers and promoters located adjacent to CTCF loop anchors deactivated during pancreatic cell differentiation (Fig. 5b and Supplementary Fig. 7b). For example, enhancers adjacent to hESC-specific loop anchors lose ATAC-TF signal indicative of dissociation of transcription factors from chromatin when cells differentiate into definitive endoderm, and levels of H3K4me1, H3K27ac, and RNAPIISer2ph decrease while levels of H3K27me3 and H3K9me3 increase (Fig. 5b and Supplementary Fig. 7b). These results suggest that changes in loop anchors are accompanied by changes in the regulatory sequences of adjacent genes.

To further examine the relationship between formation of new CTCF loops during pancreatic cell differentiation, enhancer activation, and gene expression, we used RNA-seq data and examined changes in transcription at stage-specific CTCF loops. For all stages of pancreatic cell differentiation, formation of new CTCF loops is accompanied by increases in transcription of genes contained within the loops as exemplified for loops formed in SC-β organoid cells (Fig. 5c). We then used Hi-C and HiChIP data for each differentiation stage to analyze the relationship between the formation of new CTCF loops and activation of enhancers and promoters responsible for changes in transcription

within these loops using meta APA analyses. We find that, for each stage, increases in interactions between CTCF anchors of stage-specific loops is accompanied by increases in interactions between enhancers and promoters within the same loops. For example, subtraction maps of Hi-C data from PP and H9 hESC cells showing meta APA analyses of anchored PP-specific CTCF loops indicate an increase in enhancer-promoter interactions within the loops in PP with respect to hESCs (Fig. 5d, top left). The same is true for comparisons between SC-β organoids and hESC cells (Fig. 5d, top right). Mapping RNA-PIIS2ph HiChIP signal on the same Hi-C matrix highlights the increased interactions between enhancers and promoters, although it cannot detect interactions between CTCF anchors (Fig. 5d, bottom panels).

Results from a second type of analysis also support a correlation between the formation of new CTCF loops and the activation of enhancers and promoters located in adjacent sequences. As described above, most new stage-specific CTCF loops form by the extension of one or both original loop anchors present in the previous stage. To analyze the relationship between the formation of new CTCF loops via the different strategies shown in Fig. 3b and the activation of adjacent regulatory sequences, we first selected stage-specific enhancers and promoters located within 10 kb of stage-specific CTCF sites in loop anchors. We then performed meta-analyses of H3K27ac HiChIP data with the assumption that interactions between active enhancers and promoters detected by H3K27ac HiChIP would appear to coincide with CTCF loop anchors at this level of resolution. For stage-specific CTCF loops that form via the replacement of one anchor for a second more distant anchor, the transition between consecutive differentiation stages is accompanied by increased enhancer-promoter interactions adjacent to the CTCF anchors (Fig. 5e). Similarly, when new CTCF loops are formed by utilization of more distant sites at both anchors, new enhancer-promoter interactions detected by H3K27ac HiChIP are established at each differentiation stage adjacent to new CTCF loop anchors (Fig. 5f). Together, these results suggest that new transcription patterns established during lineage specification involve the activation of new enhancers and new CTCF loops between adjacent anchors to allow the interaction of these enhancers with their cognate promoters.

## Changes in cohesin loading during pancreatic cell differentiation

Loading of the cohesin complex takes place at genomic sites different from loop anchors via NIPBL-mediated recruitment[72]. NIPBL has been shown to be enriched at active enhancers and promoters[73,74]. Therefore, as the active state of promoters and enhancers changes during cell differentiation, the genomic sites where cohesin loads may also change. If cohesin extrusion stops at the first two convergent CTCF sites it encounters, the ability of these sites to function as loop anchors will change depending on where cohesin loads. This may explain why different CTCF sites with similar levels of this protein are used at different differentiation stages. To explore this issue in detail, we selected all the loops present in DE cells but absent in H9 hESCs and we scaled all new loops to the same size. Assuming that sites of NIPBL present within a specific loop and also containing RAD21 represent cohesin loading sites from where the cohesin complex extrudes to form this loop, we then ranked the loops based on the distance between the anchors and the NIPBL/RAD21 sites internal to the loop. This approach allows the visualization of putative NIPBL loading sites within the new loops with respect to proteins of interest. We then plotted changes in CTCF, RAD21, and WAPL in the same ranking order as NIPBL. The results suggest that anchors of loops not present in H9 cells and formed in DE cells flank potential loading sites that lack RAD21, NIPBL, and WAPL in H9 hESCs but contain all three proteins in DE cells when the loop is formed (Fig. 6a). As expected, CTCF is present at the anchors shown as vertical lines in the heatmaps (Fig. 6a) in DE cells at higher levels than in H9, but it is

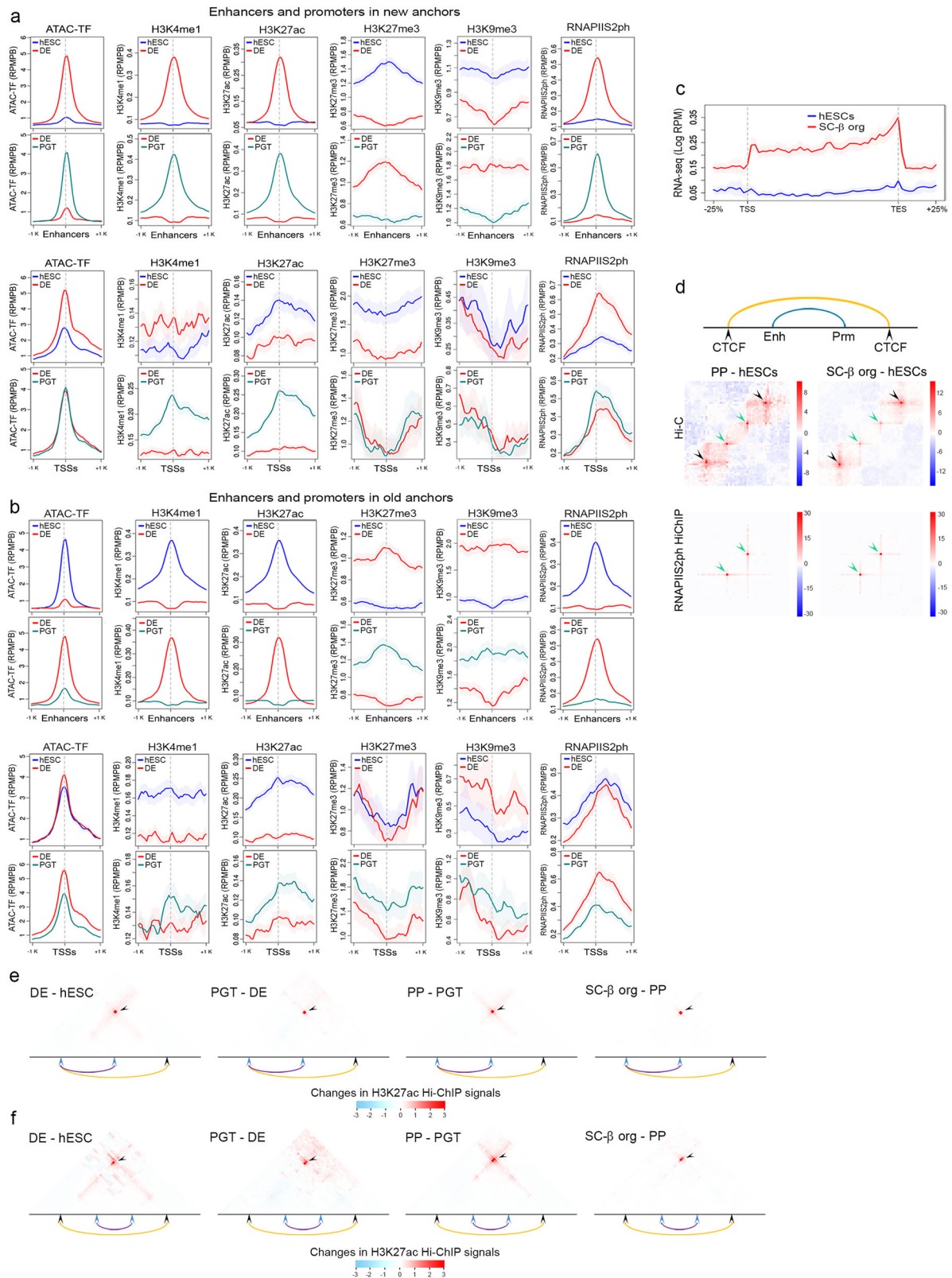

not present at loading sites present in the diagonal connecting the anchors in the Hi-C heatmaps.

To gain further insights into the nature of the NIPBL loading sites, we mapped ChIP-seq data for H3K4me1, H3K27ac, and RNPIISer2ph on the same matrix obtained with NIPBL/RAD21 sites. We find that these potential loading sites are enriched for RNPIISer2ph and these two histone modifications in DE cells but not H9, suggesting that these

regions correspond to enhancers activated during the differentiation of hESCs into definitive endoderm (Fig. 6b). Furthermore, YY1, a transcription factor enriched at promoters and tissue-specific enhancers[75], also becomes enriched at these NIPBL sites concomitant with the differentiation of DE cells (Fig. 6b). Similar results obtained at the transition of PP cells into SC-β organoids are shown in Supplementary Fig. 8a and Supplementary Fig. 8b. A specific example in

**Fig. 5 | Chromatin changes at enhancers and TSSs adjacent to new and old CTCF loop anchors. a** Formation of CTCF loops by recruiting CTCF to new anchor sites correlates with an increase in ATAC-TF signal at adjacent enhancers and transcription start sites (TSSs). These enhancers also show increased H3K4me1, H3K27ac, and RNAPIIS2ph. Promoters show similar changes over broader regions. **b** Dissolution of CTCF loops by discarding previously used anchor sites. When this happens at a specific stage, enhancers and TSSs adjacent to discarded CTCF anchor sites lose ATAC-TF signal, H3K4me1, H3K27ac, and RNAPIIS2ph, suggesting that loss of CTCF anchors correlates with inactivation of adjacent enhancers and promoters. **c** Changes in transcription measured by RNA-seq at genes located inside stage-specific CTCF loops. **d** Metaplot analysis of interactions between CTCF loop anchors and between enhancers and promoters located within the loop anchors. The top diagram indicates the specific arrangement of CTCF loop anchors and enhancer-promoter interactions analyzed in this specific case. Changes in interactions between enhancer-promoter pairs in two differentiation stages are analyzed in the context of changes in CTCF loops in which the enhancer-promoter pairs are contained. Below the diagram, the top left panel shows a subtraction heatmap of

Hi-C interactions in PP and H9 hESCs for PP-specific CTCF loops. Formation of these loops in PP cells (black arrowheads indicating increased interactions at the CTCF loop anchors) correlates with increased enhancer-promoter interactions inside the loops (green arrowheads). The right panel shows a similar analysis comparing CTCF loops present in SC-β organoids but absent in H9 hESCs. The bottom panels show a parallel analysis using RNAPIIS2ph HiChIP instead of Hi-C. Interactions at CTCF loop anchors cannot be detected (absence of black arrowheads) because CTCF anchors lack RNAPIIS2ph; however, enhancer-promoter interactions can be observed as increased signal in RNAPIIS2ph HiChIP data (green arrowheads) **e** Subtraction heatmaps of H3K27ac HiChIP between consecutive differentiation stages around new CTCF loop anchors formed by extension of one old anchor. **f** Subtraction heatmaps of H3K27ac HiChIP between consecutive differentiation stages around new CTCF loop anchors formed by extension of two old anchors. Abbreviations: human embryonic stem cells (hESCs), definitive endoderm (DE), primitive gut tube-like (PGT), pancreatic progenitors (PP), and stem cell-derived β-cell organoids (SC-β organoids).

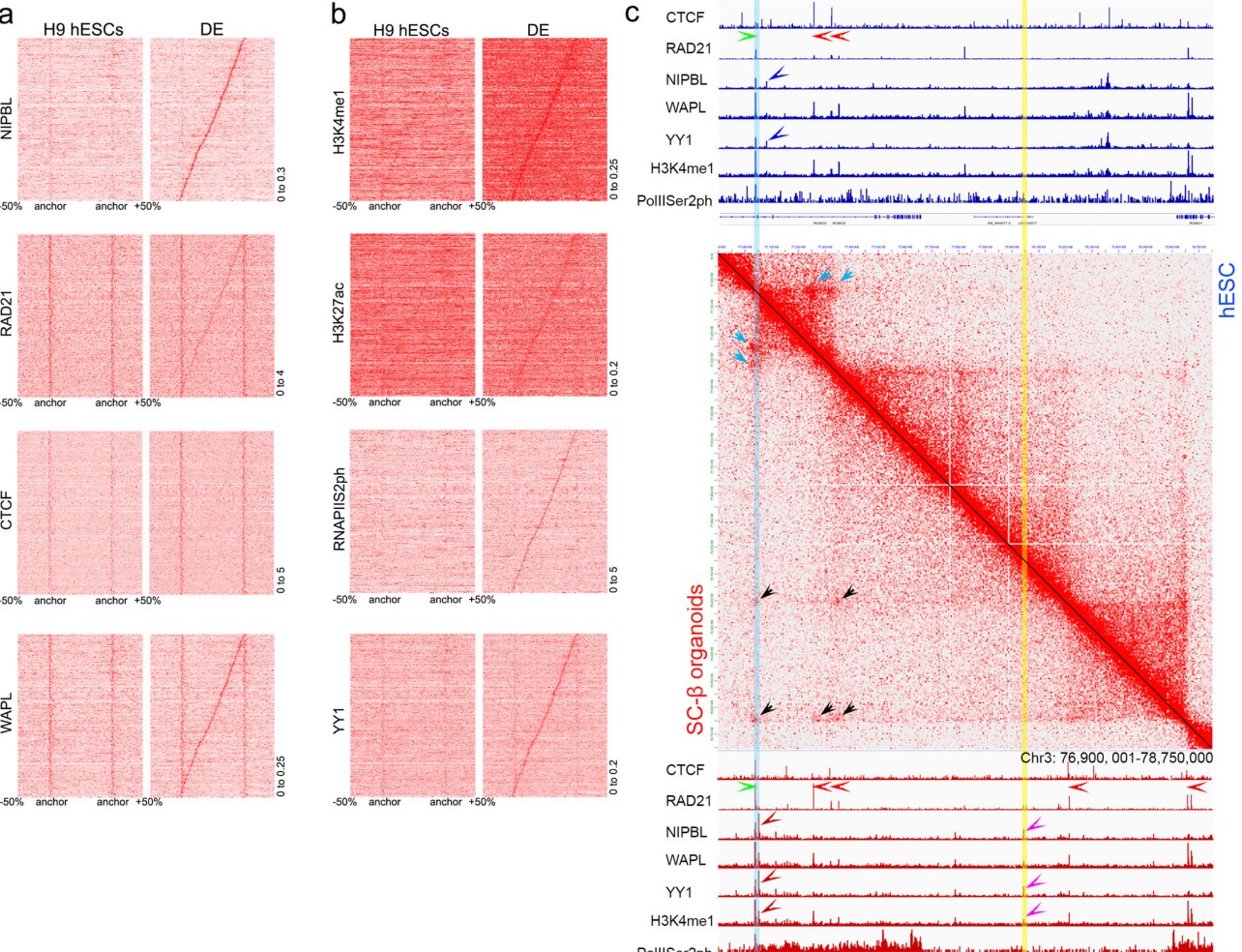

**Fig. 6 | Distribution of various proteins at CTCF loop anchors and cohesin loading sites. a** Distribution of NIPBL, RAD21, CTCF and WAPL at loop anchors and loading sites of CTCF loops identified from Hi-C data present in DE cells but not H9 hESCs. **b** Distribution of H3K4me1, H3K27ac, RNAPIIS2ph, and YY1 at loop anchors and loading sites of CTCF loops identified from Hi-C data present in DE cells but not H9 hESCs. **c** Example of a specific genomic region representative of the metaplots shown in (**a**, **b**). The top section shows the distribution of several proteins and histone modifications in this region of the genome in H9 hESCs. Green and red arrowheads indicate orientation of the CTCF motif at the site. Blue arrowheads show possible sites of cohesin loading to make the CTCF loops observed in Hi-C

data. The middle section shows Hi-C heatmaps in H9 hESCs and SC-β organoids. Blue arrowheads show CTCF loops present in both cell types whereas black arrowheads show those found only in SC-β organoids. The bottom section shows the distribution of several proteins and histone modifications in the same genomic region in SC-β organoids. Pink and dark red arrowheads show the location of NIPBL/ YY1 sites present in SC-β organoids but not in H9 hESCs. Abbreviations: human embryonic stem cells (hESCs), definitive endoderm (DE), primitive gut tube-like (PGT), pancreatic progenitors (PP), and stem cell-derived β-cell organoids (SC-β organoids).

which the formation of new loops correlates with the presence of a new NIPBL site in the region between the new loop anchors is shown in Fig. 6c. These results suggest that, during cell differentiation, activation of enhancers may be accompanied by the use of these enhancers as loading sites for the cohesin complex. If the target promoter of an enhancer contains a CTCF site in the appropriate orientation, this strategy ensures that the enhancer is targeted to the appropriate gene.

## Discussion

Observations in humans and laboratory animals have uncovered a variety of tissue-specific phenotypes caused by CTCF mutations[76,77]. These phenotypes indicate a role for CTCF in the differentiation of various cell types during development, suggesting a requirement for CTCF-mediated regulation of enhancer-promoter interactions during cell fate specification. Here, we examine the differentiation of hESCs into pancreatic cells as a model to explore the mechanisms by which CTCF loops are established and dissolved at specific developmental stages and their effect on enhancer function. TADs are normally identified using algorithms that detect changes in the directionality of interactions along the genome. Instead, we identified only CTCF loops using algorithms that detect corner dots, the punctate intense signal observed at the summits of a subset of domains in Hi-C heatmaps[3,51]. By comparing CTCF loops identified in H9 hESCs, DE, PGT, PP, and SC-β organoids we find that rather than being constant, CTCF loops are very dynamic during cell differentiation, forming and disassembling during developmental transitions to create stage-specific loops. These developmentally dynamic CTCF loops are formed by at least two different mechanisms. One involves the recruitment of CTCF to new, previously unoccupied, sites in the genome. The second involves the use of CTCF sites that are occupied throughout the differentiation process but are only used as anchors at specific stages.

Unoccupied CTCF sites at any given stage are located within large DNA methylation domains and short regions containing variable levels of H3K9me3 and the ChAHP complex. De novo occupancy by CTCF at specific stages of pancreatic cell differentiation correlates with ejection of at least the CHD4 component of ChAHP—other components were not tested in these studies—demethylation of H3K9me3 into H3K9me2, and local DNA demethylation of an approximately 600 bp region surrounding the CTCF site. Additional changes that correlate with recruitment of CTCF to these sites include the recruitment of pioneer factors and positioning of nucleosomes flanking the new sites. At this time, we are unable to distinguish the relative timing of these events or their causal role in allowing CTCF to bind to these previously unoccupied sites. It is important to consider that these events not only lead to the eventual formation of new CTCF loops but also to changes in compartmental interactions with other genomic regions containing H3K9me3, which presumably form biomolecular condensates. Sequences surrounding the new CTCF anchors that contained H3K9me3 in the previous stage, now contain H3K9me2, and fail to interact with other regions of the genome through compartmental interactions[10,78].

Interestingly, many new CTCF loops that form at each differentiation stage do so by using anchors that were occupied by CTCF at the previous stage. When these loops disassemble in a subsequent stage, they do so without significant alterations in CTCF occupancy. Assuming an average of 60,000 occupied CTCF sites in the human genome[79], there is one CTCF site every 50 kb, approximately. The median size of a CTCF loop is 360 kb[62], suggesting that many CTCF sites in the genome do not serve as loop anchors in a specific cell type. If cohesin extrusion stops at the first pair of CTCF convergent sites it encounters, one strategy for cells to regulate the use of CTCF sites as loop anchors would be to regulate the loading sites for the cohesin complex. This appears to be the case during the differentiation of hESCs into pancreatic SC-β organoids, when a subset of loops present at a specific differentiation stage are formed between pre-existing CTCF sites that were not forming loops in the previous stage. The formation of these loops correlates with de novo recruitment of NIPBL to sites located between the new anchors. These sites are enriched in H3K4me1, H3K27ac, RNA-PIIS2ph, and YY1, suggesting that they correspond to enhancers activated at the same developmental stage. It is unclear whether YY1 is simply a transcription factor present at these enhancers for the purpose of mediating enhancer function or whether it is present at these sites to cooperate with NIPBL and load the cohesin complex. These observations suggest a coordination between the activation of enhancers and of CTCF loop anchors. A strategy by cells of loading cohesin at active enhancers would ensure that these enhancers are within the same loop as their target promoters. Since many gene promoters contain CTCF sites, this strategy would also result in the positioning of the enhancer in close proximity to the first promoter cohesin encounters containing a CTCF site with the opposite orientation to the direction of extrusion.

## Methods

### Cell lines and culture

The human embryonic stem cell line H9 (WA09) was obtained from WiCell and was utilized for differentiation of pancreatic cells. Cells were cultured in STEMPRO hESC SFM (Thermo Fisher, A1033201) on cultureware coated with Geltrex Matrix (Thermo Fisher, A1569601) at 37 °C under 5% $CO_2$. Medium was changed every day. To induce pancreatic differentiation, H9 cells were seeded at $1 \times 10^6$ cells/ml in 5.5-6 ml of mTeSR1 media (STEMCELL Technologies, 85857) with 10 μM of Y27632 (Selleckchem, S1049) per well of a 6-well spheroid plate (Greiner, 657970), and incubated on an Innova 2000 rotator at 97 rpm, 37 °C under 5% $CO_2$ overnight to form spheroids. The spheroids were fed with fresh mTeSR1 with Y27632 after 24 h and 48 h. After 72 h, the spheroids were collected and a stepwise differentiation was started by changing the medium supplied with different ingredients as described below:

Day 1: S1 + 100 ng/ml Activin A (R&D Systems, 338-AC) + 3 μM CHIR99021 (Selleckchem, S2924) + 10 μM Y27632. Day 2: S1 + 100 ng/ml Activin A. Days 3, 5, 10, 12, 15, 17 and 19, no feed. Days 4, 6: S2 + 50 ng/ml KGF (Peprotech, AF-100-19). Days 7, 8: S3 + 50 ng/ml KGF + 0.25 μM Sant1 (Sigma, S4572) + 2 μM RA (Sigma, R2625) + 200 nM LDN193189 (only Day 7) (Sigma, SML0559) + 500 nM PdBU (EMD Millipore, 524390) + 10 μM Y27632. Days 9, 11, 13: S3 + 50 ng/ml KGF + 0.25 μM Sant1 + 100 nM RA + 10 μM Y27632. Days 14, 16: S5 + 0.25 μM Sant1 + 100 nM RA + 1 μM XXI (Sigma, 565790) + 20 ng/ml Betacellulin (Peprotech, 100-50) + 10 μM Alk5i II (Enzo, ALX-270-445-M001) + 1 μM T3 (Sigma, 64245-250MG-M). Days 18, 20: S5 + 20 ng/ ml Betacellulin + 1 μM XXI + 10 μM Alk5i II + 1 μM T3 + 25 nM RA. Days 21−35: S6 + 10 μM Alk5i II + 1 μM T3 with media changes every second day. Cells differentiated into DE, PGT, PP and SC-β organoid stages were collect at Day 4, Day 7, Day 14, and Day 35, respectively.

### FACS of fixed cells

Cultured spheroids were dissociated to single cells using Accutase (Thermol, A1110501) and fixed with 4% paraformaldehyde in PBS for 30 min at room temperature before flow cytometry. Fixed cells were permeabilized and blocked in PBS with 5% donkey serum (Jackson Immunoresearch) and 0.15% Triton X-100 (Sigma) for 20 min at room temperature. Cells were then stained with primary antibodies diluted with blocking buffer at 4 °C overnight. After staining, cells were washed, incubated with appropriate secondary antibodies for 30 min at room temperature and then resuspended in FACS buffer for flow acquisition and analysis. Cells were filtered through a 40 μm nylon mesh (BD Biosciences) and loaded on a FACScanto (BD Biosciences) for flow cytometry analysis using FlowJo software (TreeStar). Antibodies used for intracellular flow cytometry are C-peptide (DSHB; GN-ID4), Glucagon (R&D; MAB1249-SP), Chicken anti-Mouse IgG (H + L)

Cross-Adsorbed Secondary Antibody (Alexa Fluor 594), and Goat Anti-Rat IgG H&L (Alexa Fluor 488).

## Immunohistochemistry

Organoids were fixed with 4% PFA for at least 30 min at 4 °C followed by addition of 30% (wt/vol) sucrose to facilitate cryoprotection for 24-48 hr. Subsequently, organoids were stained with methylene blue and embedded with Cryo O.C.T. on dry ice. After completely frozen, organoids were sectioned in slices of 10 to 15 μm thickness. Slides were blocked in PBS, 0.3% Triton X-100 (VWR; EM-9400), and 10% donkey serum (Jackson Immunoresearch; 017-000-121) for 15 min at room temperature in a humidified chamber and stained with primary antibodies diluted with blocking buffer overnight at 4 °C. After several washes, slides were stained with secondary antibodies diluted in blocking solution for 1 h at room temperature in a humidified chamber, washed twice, and covered with VECTASHIELD Antifade Mounting Medium with DAPI (Vector Laboratories, H-1200). Slides were observed under a fluorescence microscope. The primary antibodies used include SOX17 (R&D Systems; AF1924) for DE cells; PDX1 (R&D Systems; AF2419) and NKX6.1 (DSHB; F55A12) for PP cells; and C-peptide (DSHB; GN-ID4) and Glucagon (R&D Systems; MAB1249-SP) for SC-β organoids. The secondary antibodies used include chicken anti-Mouse IgG (H + L) Cross-Adsorbed Secondary Antibody (Alexa Fluor 594), goat Anti-Rat IgG H&L (Alexa Fluor 488), and donkey anti-Goat IgG (H + L) Cross-Adsorbed Secondary Antibody (Alexa Fluor 488).

## RNA-seq

Total RNA was isolated from H9 cells and the different in vitro differentiated stages using Trizol reagent (Invitrogen) and ribosomal RNA was removed using the Ribo Minus Transcriptome isolation kit (Invitrogen, K1550). RNA concentration was measured using the Qubit RNA HS Assay kit (Thermo Fisher) and fragmented randomly by adding fragmentation buffer. cDNA was synthesized using the RNA template and random hexamer primers. After terminal repair, A ligation, and sequencing adaptor ligation, the double-stranded cDNA library was completed by size selection and PCR enrichment. Two independent biological replicates per sample were then sequenced using paired-end 50 bp on an Illumina NovaSeq 6000 instrument.

## In situ Hi-C

In-situ Hi-C libraries were prepared using DpnII restriction enzyme as previously described[51]. Briefly, 2.5 million cells at each differentiation stage were crosslinked with 1% formaldehyde, quenched with glycine, washed with PBS, and permeabilized to obtain intact nuclei. Nuclear DNA was then digested with DpnII, the 5′-overhangs were filled with biotinylated dCTPs and dA/dT/dGTPs to make blunt-end fragments, which were then ligated, reverse-crosslinked, and purified by standard DNA ethanol precipitation. Purified DNA was sonicated to 200−500 bp fragments and captured with streptavidin beads. Standard Illumina TruSeq library preparation steps, including end-repairing, A-tailing, and ligation with universal adaptors were performed on beads, washing twice in Tween Washing Buffer (5 mM Tris-HCl pH 7.5, 0.5 mM EDTA, 1 M NaCl, 0.05% Tween 20) between each step. DNA was PCR amplified on the beads with barcoded primers using KAPA SYBR FAST qPCR Master Mix (Kapa Biosystems) for 5−12 PCR cycles to obtain enough DNA for sequencing. Libraries were paired-end sequenced on an Illumina NovaSeq 6000 instrument. Two biological replicates were generated, and replicates were combined for all analyses after ensuring high correlation.

## HiChIP

HiChIP libraries were prepared using H9 and differentiated cells at different stages as described[80] with some modifications. Cells were collected, fixed with 1% formaldehyde for 10 min at room temperature, quenched with glycine, washed with PBS and stored frozen at −80 °C. For library preparation, fixed cells were gently homogenized in Hi-C lysis buffer with pestle A to release nuclei, followed by DpnII digestion, biotin-dCTP fill in, and re-ligation. After ligation, chromatin was sheared by sonication into 200-500 bp fragments and precleared with Protein A and G dynabeads at 4 °C for 2 h, then precipitated using Protein A or G dynabeads and pre-incubated with appropriate antibodies overnight to enrich for ligation products bound by specific proteins. Tagmentation of immunoprecipitated chromatin with Tn5 transposase mixture was performed on beads. After elution, reverse crosslinking and ethanol precipitation, a second pull down with streptavidin beads was performed to enrich for biotin-labeled ligation products. On bead PCR amplification was performed to derive libraries for sequencing on an Illumina NovaSeq 6000 instrument. Two replicates of each sample were obtained.

## MiChIP

We also performed a variation of HiChIP, which we will refer to as MiChIP, in which the initial fixation steps and chromatin digestion were performed as in Micro-C[51]. H9 cells and in vitro differentiated cells at different stages were collected, sequentially fixed with 3 mM DSG for 40 min and 1% formaldehyde for 10 min at room temperature, quenched with 0.2 M glycine for 5 min at room temperature, washed with PBS and stored frozen at −80 °C. For library preparation, fixed cells were gently homogenized in MB#1 (10 mM Tris-HCl, pH 7.5, 50 mM NaCl, 5 mM MgCl$_2$, 1 mM CaCl$_2$, 0.2% NP-40, 1x Roche cOmplete EDTA-free) with pestle A. Chromatin was fragmented with MNase for 10 min at 37 °C and digestion was stopped with 5 mM EGTA at 65 °C for 10 min. The chromatin was resuspended in 1x NEBuffer 2.1 (NEB, #B7202S) and dephosphorylated by the addition of 5 μl rSAP (NEB, #M0203) at 37 °C for 45 min. 5′ overhangs were generated with the following pre-mix (50 mM NaCl, 10 mM Tris, 10 mM MgCl$_2$, 100 μg/ml BSA, 2 mM ATP, 3 mM DTT, 8 μl Large Klenow Fragment (NEB, #M0210L) and 2 μl T4 PNK (NEB, #M0201L) at 37 °C for 15 min. The DNA overhangs were filled with biotinylated nucleotides by the addition of 100 μl pre-mix (25 μl, 0.4 mM Biotin-dATP (Invitrogen, #19524016), 25 μl 0.4 mM Biotin-dCTP (Invitrogen, #19518018), 2 μl 10 mM dGTP and 10 mM dTTP (stock solutions: NEB, #N0446), 10 μl 10x T4 DNA Ligase Reaction Buffer (NEB #B0202S), 0.5 μl 200x BSA (NEB, #B9000S), 38.5 μl H$_2$O) and incubated at 25 °C for 45 min. The reaction was stopped by the addition of 12 μl 0.5 M EDTA (Invitrogen, #15575038) at 65 °C for 20 min. After proximity ligation and removal of unligated ends, chromatin was sheared by sonication into 200-500 bp fragments and precleared with Protein A and G Dynabeads at 4 °C for 2 h, then precipitated using Protein A or G dynabeads pre-incubated with appropriate antibodies overnight to enrich for ligation products bound by specific proteins. Immunoprecipitated ligation products were eluted, reverse crosslinked and ethanol precipitated. Purified DNA was further cleaned with Dynabeads™ MyOne™ Streptavidin C1 beads (Invitrogen, #65001) to enrich for biotin-labeled ligation products. After end-repair, A-tailing and Truseq adapter ligation, PCR amplification was finally performed on beads using Truseq barcoded primers to generate libraries for sequencing on an Illumina NovaSeq 6000 instrument.

## ChIP-seq

ChIP-seq experiments were carried out as follows[82]. After removal of medium, cells were crosslinked in 1% formaldehyde in PBS at room temperature for 10 min and quenched with glycine. PBS-rinsed cell pellets were flash frozen in liquid nitrogen and stored at -80 °C or used immediately. After cell lysis, chromatin was sonicated into 200-500 bp fragments, precleared with Protein A or G Dynabeads at 4 °C for 2 h, and precipitated with antibody overnight at 4 °C. Immunoprecipitated chromatin was tagmented with Tn5 transposase mixture on beads and then eluted, reverse crosslinked and purified by standard methods.

Purified DNA was amplified with Illumina Nextera barcoded primers using KAPA SYBR FAST qPCR Master Mix for 5 - 12 PCR cycles to obtain enough DNA for sequencing.

## ATAC-seq

H9 cells and in vitro differentiated cells at different stages were collected and used to perform ATAC-seq[83,84]. SC-β organoids were homogenized with an electric homogenizer for 10 seconds into small clusters of cells. Cells were washed with PBS and the nuclear membrane was disrupted by soaking in lysis buffer (10 mM Tris-HCl pH 7.4, 10 mM NaCl, 3 mM $MgCl_2$, 0.1% NP40, 0.1% Tween-20, and 0.01% Digitonin) on ice. Nuclei were washed once in cold lysis buffer without NP40 or digitonin and then incubated in Tn5 transposase mixture (25 µl 2x TD buffer, 2.5 µl Tn5 (100 nM final), 16.5 µl PBS, 0.5 µl 1% digitonin, 0.5 µl 10% Tween-20, 5 µl $H_2O$) at 37 °C for 20 min with occasional shaking. After the reaction was completed, DNA was extracted using the Minelute Kit (Qiagen). Purified tagmented DNA was PCR amplified and sequenced in a Novaseq 6000 instrument.

## CUT&Tag

H9 cells and in vitro differentiated cells at different stages were harvested and gently homogenized with pestle A. Cells were then washed twice in 1.5 mL Wash Buffer (20 mM HEPES pH 7.5, 150 mM NaCl, 0.5 mM Spermidine, 1 × Protease inhibitor cocktail, EDTA free), immobilized to concanavalin A coated magnetic beads (Bangs Laboratories), and then resuspended in 50 µl Dig-wash Buffer (20 mM HEPES pH 7.5, 150 mM NaCl, 0.5 mM Spermidine, 1 × Protease inhibitor cocktail, 0.05% Digitonin) containing 2 mM EDTA[85]. After sequential incubation with antibodies to CHD4 (Abcam, #ab72418, diluted 1:50 in 50 µl of Dig-Wash buffer), secondary antibodies (diluted 1:100 in 100 µl of Dig-Wash buffer) and a 1:200 dilution of pAG-Tn5 (gift from S. Henikoff) in Dig-300 Buffer (0.05% digitonin, 20 mM HEPES, pH7.5, 300 mM NaCl, 0.5 mM Spermidine, 1 × Protease inhibitor cocktail), bead-bound cells were resuspended in 50 µl tagmentation buffer (10 mM $MgCl_2$ in Dig-wash Buffer). The tagmented DNA was cleaned with 1.5 × Ampure XP beads (Beckman Counter), amplified with Illumina Nextera barcoded primers, and purified by 1.1 × Ampure XP beads for sequencing in a Novaseq 6000 instrument.

## Data processing

**Analysis of ChIP-seq and CUT&Tag data.** All reads were mapped to unique genomic regions using Bowtie2 and the hg38 human genome release. PCR duplicates were removed using Picard Tools (http://picard.sourceforge.net; https://broadinstitute.github.io/picard/). The Bedtools Genome Coverage function was used to derive bedgraph files for further analysis. To compare changes in ChIP-seq signals, libraries were normalized by random picking to obtain the same numbers of mapped reads. Normalized reads were used to derive bedgraph files for comparison in IGV. MACS2 was used to call peaks using default parameters with IgG ChIP-seq data as control. Differential peaks were found using the edgeR[86] R package at $p < 0.05$, changes over 20% up/down. H3K9me3 changes during differentiation were evaluated in 25 kb bins where positive H3K9me3 bins had an IP / Input signal >= 4 and differential bins were identified by at least a four-fold change. Significant TF motifs present at ChIP-seq peaks and at differential loop anchors were found using MEME. Exact motif sequences were scanned using FIMO and the JASPAR_CORE_2016_vertebrates database against a set of peaks or anchors to obtain the overlapping percentages.

**RNA-seq data processing.** RNA-seq raw reads were aligned using HISAT2 v2.2.0 to the hg38 human genome with default parameters. Transcripts per million (TPM) counts for all annotated human genes and transcripts were calculated using StringTie v2.1.6. Differentially expressed genes were identified using the R package edgeR with a cutoff $p$-value ≤ 0.05 and fold change over 20% up/down.

**Hi-C data processing.** Paired-end reads from Hi-C experiments were aligned to the human hg38 reference genome using HiC-Pro v2.10.0. After removal of PCR duplicates and low-quality reads, high-quality reads were assigned to DpnII restriction fragments, filtered for valid interaction contacts, and used to generate binned contact matrix hic files[87,88]. For visualization and further analysis of Hi-C contact maps, Knight-Ruiz (KR) normalized signal for the interaction matrices were derived using the Juicebox tools dump command[87]. SIP v1.3.3 (https://github.com/PouletAxel/SIP/releases) was used to call CTCF loops in the Hi-C interaction matrix[52,88]. Fit-Hi-C (https://github.com/ay-lab/fithic)[89] was used to call significant interactions at 5, 10, and 25 kb resolution with a q value cutoff of > 0.001 and merged together for analyses.

**HiChIP and MiChIP data processing.** Paired-end reads from HiChIP and MiChIP experiments were aligned to the human hg38 reference genome using HiC-Pro v2.10.0. After PCR duplicates and low-quality reads were removed, high-quality reads were assigned to DpnII restriction fragments, filtered for valid interaction contacts, and used to generate binned contact matrix hic files. For visualization and further analysis of HiChIP and MiChIP contact maps, vanilla coverage square root (VCsqrt) normalized signal for the interaction matrices were derived using the Juicebox tools dump command[87]. FitHiChIP (https://ay-lab.github.io/FitHiChIP/)[90] was used to generate singleton reads resembling ChIP-seq data for finding genomic targets of specific proteins, and to call significant interactions with default parameters with an FDR cutoff at 0.05 for finding long-range contacts associated with specific proteins.

**Hi-C and HiChIP analysis.** Hi-C and HiChIP contact matrices were processed using the Juicer pipeline[88]. For downstream analysis, matrices were distance normalized via the formula (observed−expected)/(expected + 1). Comparison of Hi-C or HiChIP was done on distance normalized reads from matrices randomly sampled to contact the same total Hi-C contacts between samples. Traditional A/B compartments were identified through the eigenvector of the Pearson correlation matrix at 25 kb resolution as described[91]. Candidate CTCF loops in each sample were identified using SIP[52] at 5 kb and 10 kb resolution from which a total master list of potential loops in any sample was created. Loop calling parameters for SIP were as follows: -norm KR -min 2.0 -max 2.0 -mat 2000 -d 6 -res 5000 -sat 0.01 -t 2500 -nbZero 6 -factor 1 -fdr 0.05 -del true -cpu 48 -factor 4. For comparisons between different Hi-C libraries, the following normalization steps were taken, (1) valid contacts from each library were randomly picked to match the size of the library with the lowest numbers of contacts; (2) KR normalization was applied to obtain the balanced matrices. (3) Matrices were then distance normalized by the formula (observed −expected)/(expected + 1). The normalized matrices were then used to call differential loops of all stages using the following approach: (1) loops obtained using SIP from all stages were combined; (2) KR and distance normalized contact frequencies in all stages were combined pairwise on all resulting combined loops in step 1; (3) pairwise interaction frequencies from step 2 were used as input in the edgeR R package to identify significant differential loops for each stage (FDR cutoff <0.1, $p$-value < 0.05 and fold change ≥4). To further identify stage specific loops, the following steps were taken, (1) all differential loops between all stages were combined; (2) contact frequencies were calculated for all stages; (3) loops were ranked by the stages when their contact frequencies reach the maximum (for gained loops) or minimum value (for lost loops); (4) loops were allocated to each stage when they reach maximum changes based on step 3 and defined as stage specific loops. To find common loops, combined SIP loops with FDR cutoff ≥ 0.1, $p$-value≥ 0.05 or fold change < 4 in edgeR were excluded from stage specific differential loops and defined as common loops. Metaplots of loops and the surrounding 100 kb were calculated

using SIPMeta with Manhattan distances[52]. Meta scores were calculated by the intensity of the center bin divided by the median signal of the four bins in the top right corner, similar to APA analysis[51]. Changes to interactions in the proximity of the loop were calculated by measuring differences in average signal in metaplots for the top left corner (category 1, inside-outside left of loop), the bottom right corner (category 2, inside-outside right of loop), and the top right corner (category 3, crossing over the loop). Motifs enriched in the anchors of increased loops were identified by MEME-ChIP using the summits of overlapping ATAC-seq peaks. Profiles across motifs were performed by randomly sampling reads to have the same number between samples and using ngsplot. Significant interactions were obtained via Fit-Hi-C for Hi-C data and FitHiChIP for HiChIP or MiChIP data in 10 kb bins.

**APA metaplot analysis.** Aggregate peak analysis (APA) metaplots and scores were generated as described[51] using 10 kb resolution contact matrices. To measure the enrichment of loops over the local background and normalize for different loop distances and protein occupancy bias, we collected the VCsqrt normalized observed over expected (O/E) contact frequency of pixels of loops as well as the surrounding pixels up to 10 bins away in both x and y directions i.e., 210 kb*210 kb local contact matrices. The median O/E for each position of all 210 kb*210 kb contact matrices for a set of loops were calculated and plotted using the heatmap.2R package to generate the aggregate heatmaps. APA scores were determined by dividing the center pixel value by the mean value of the 25 (5*5) pixels in the lower right section of the APA plot.

**Multiple anchor metaplots.** Multiple anchor metaplots were obtained at 10 kb resolution and the distance between anchors was scaled to 10 equal bins. For three CTCF anchors, the anchors were oriented such that the stable anchors are the first ones on the left. To compare libraries from cells subjected to different treatments, the observed interaction matrices were normalized between samples by random picking to obtain equal numbers of contacts for each library. VCsqrt normalization was then applied to all contact matrices. To compare HiChIP aggregate signal changes between different samples on distinct anchor sets, subtraction or log2 fold changes of treatment versus control were calculated for each loop separately and then summarized by taking the median values of all anchors for visualization.

**Overlapping of ChIP-seq peaks with HiChIP loop anchors.** For analyzing overlaps between ChIP-seq peaks and loop anchors, Mango[92] loop anchors ± 5 kb were used to overlap with ChIP-seq peaks using the bedtools intersect function. ChIP-seq peaks were shuffled 1000 times and the same analysis was repeated to obtain the expected overlapping ratio. Significant p-values were derived by numbers of times when observed <expected happens divided by 1000.

**Enhancer definition.** Enhancers were defined by using H3K4me1 peaks without H3K4me3 but overlapping ATAC-seq-TF peaks and excluding TSS ± 1 bp. Among these enhancers, those overlapping H3K27ac peaks were defined as active enhancers. Differential enhancers were found using the edgeR R package based on H3K27ac ChIP-seq signals with a p-value cutoff at 0.05 and more than 3-fold changes in either condition.

**Heatmaps and average profiles of ChIP-seq and clustering.** For deriving heatmaps of ChIP-seq signal, anchors plus flanking regions were binned equally to get a blank matrix (anchors × bins). To compare between samples, reads from the same antibody analysis were normalized by random pick. Normalized read pairs were mapped to each genomic bin with the bedtools intersect function to obtain read counts in each bin for the whole matrix. To normalize

for sequencing depth, values in the matrix were divided by library sizes in millions to obtain reads per million per covered bin (RPMPCG or RPM), and the result was then visualized with Java treeview to derive heatmaps. Average profiles of the ChIP-seq or ATAC-seq data were calculated and plotted using mean values of bins at the same distances from specific anchors. K-means clustering of ChIP-seq heatmaps was done using Cluster3 on center ± 3 bins signals of the appropriate heatmaps.

**ATAC-seq data processing.** ATAC-seq data was processed using an in-house pipeline. First, paired end reads were aligned to the hg38 human reference genome using Bowtie2 with default parameters except -X 2000 -m 1. PCR duplicates were removed using Picard Tools (http://picard.sourceforge.net; https://broadinstitute.github.io/picard/). To adjust for fragment sizes, reads mapped to + strands and -strands were offset by +4 bp and -5 bp respectively. For all ATAC-seq datasets, sub-nucleosome and mono-nucleosome reads were separated based on fragment sizes ATAC-TF 50-115 bp and ATAC-Nuc 180–247 bp. To obtain the exact positioning of nucleosomes, DANPOS[93] was used to derive the nucleosomal signals genome wide using the dpos function and 180–247 bp fragments as input and 115 bp fragments as background using -p 1 -a 1 -jd 20 -u 0 -m 1. Reads were normalized between samples before running DANPOS. Bedgraphs were made using the bedtools genomeCoverage function. MACS2 was used to call peaks for ATAC-TF reads, which are bound by transcription factors. Heatmaps and average profiles of ATAC-TF and ATAC-Nuc signals were derived as described for ChIP-seq data. Clustering of ATAC-seq heatmaps was done using Cluster3 with K-means clustering. To analyze the footprints of TFs in ATAC-TF data, motifs on a set of peaks or loop anchors were used as anchors for running dnase_average_profile.py scripts of the Wellington program in ATAC-seq mode. To compare between samples, read-normalized ATAC-TF fragments were used as input to obtain the footprint average profiles. The footprint p-values of all motifs on a set of peaks or anchors were derived using the wellington_footprints.py scripts of the Wellington program in ATAC-seq mode on read-normalized ATAC-TF fragments.

**Reporting summary**
Further information on research design is available in the Nature Portfolio Reporting Summary linked to this article.

## Data availability

All data sets generated in this study are deposited in the NCBI Gene Expression Omnibus (GEO; https://www.ncbi.nlm.nih.gov/geo/) under the following accession numbers. RNA-seq, ATAC-seq, ChIP-seq and CUT&Tag data are available under accession number GSE211101. Hi-C data is available under accession number GSE210524. HiChIP and MiChIP data are available under accession number GSE210525. The various datasets reported in the manuscript can be visualized in the UCSC browser using the following link https://genome.ucsc.edu/cgi-bin/hgTracks?db=hg38&lastVirtModeType=default&lastVirtModeExtraState=&virtModeType=default&virtMode=0&nonVirtPosition=&position=chr2%3A25160915%2D25168903&hgsid=1711140546_4oiO7H0pztEcMSrKRlpnSCYhbDET.

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

## Acknowledgements

The authors would like to thank Dr. Cynthia Vied at the Translational Science Laboratory of Florida State University for help with Illumina sequencing. This work was supported by U.S. Public Health Service Awards R35 GM139408 (VGC), R00 GM127671 and R35 GM147467 (MJR), and 5P01 GM085354 (SD) from the National Institutes of Health. The content is solely the responsibility of the authors and does not necessarily represent the official views of the National Institutes of Health.

## Author contributions

X.L. and V.G.C. designed the project and wrote the manuscript. X.L. performed all experiments. X.L. and M.J.R. performed data analyses. M.K. and S.D. planned and performed cell differentiation experiments.

## Competing interests

The authors declare no competing interests.
