## [Peer Review File · Nature Communications]

Regulation of CTCF loop formation during pancreatic cell differentiationREVIEWER COMMENTS

Reviewer #1 (Remarks to the Author):

In their manuscript Regulation of CTCF loop formation during pancreatic cell differentiation, Victor Corces and colleagues analyze the dynamics of loop formation and histone modifications during directed differentiation of human ES cells (H9) to pancreatic beta cells in-vitro. The authors show that loops may be gained between anchors binding CTCF at more immature stages of differentiation. They report that in such cases, loop formation coincides with an enhanced loading of cohesins (inferred through mapping of Nipbl) and YY1 factor inside loop domains. The authors find that the setup of new loop anchor is often related to loss of H3K9me3 and DNA methylation and gain of CTCF binding. They relate enhanced architectural loop (implicating CTCF) formation with gain of promoter-enhancer contacts between elements inside loop domains.

The authors took advantage of in-situ Hi-C to address the changes in chromatin topology, the libraries were sequenced to an ample degree allowing them to focus on changes in architectural loop formation. There are several more cutting-edge technologies used in the paper, all which allow the authors to monitor how chromatin landscape changes during pancreatic cell specification.

The paper is well written and reveals novel data. The Hi-C profiles of in-vitro differentiating human ES cells to pancreatic cells are a valuable resource in and of its own. But the paper is purely descriptive which somewhat limits its originality (or perhaps should be centered on novel findings and not on a simple description of changes in loop formation). Indeed, it seems that the novel finding would consist in the discovery of a possible link between loss of H3K9me3 and increased CTCF binding which in turn might lead to gain of chromatin loop formation. The authors also suggest that pioneer transcription factors could guide binding of CTCF to new sites in the genome.

Overall, the paper would benefit greatly from more sophisticated computational routines. It would be extremely beneficial to address differential loop formation using a statistically principled approach. Instead, the authors used a rather enigmatic approach with thresholds without any sound motivation: "Differential loops were identified where at least one sample had a distance normalized Hi-C value ≥ 4 and where there was a least a 4-fold change compared to H9 cells". How were they further filtered to uncover stage specific loops?

There are several analyses that should be included to show more convincingly the relationship between histone modifications, CTCF binding and loop formation – more details below.

The number 29,905 appears very large provided the sequencing depth, how were the common loops identified and what is the variation in signal strength at these interactions across differentiation timeline? (The word approximately (line 138) should be removed unless the number 29,905 is somewhat imprecise.)

The identification of stage specific loops should be showed more convincingly, the APA profiles should be backed up with histograms showing the distribution of fold changes of loop signal for all loop classes as some profiles of the stage specific loops show substantial Hi-C contacts in the other stages. The authors should also show the changes in loop signal without distance normalization. Such analysis will show beyond doubt that the interactions are genuinely stage specific.

There are several analyses that should be done to clarify the relationship between chromatin structure and H3K9me3.

- It is unclear how, in Figure 1, the distinction between loops with high and low level of H3K9me3 was made. The profiles look rather noisy and do not show a clear and uniform gain of H3K9me3. The authors should display $\log_2(\text{fold change})$ of H3K9me3 at loops. How does the plot of H3K9me3 signal

against CTCF binding at loop anchors look like? Can one distinguish clear populations of H3K9me3+ and H3K9me3- CTCF anchors? How reproducible are these effects? The authors should show the Hi-C signal for these loops in two replicates of their Hi-C data for each stage separately. What would be the assignment of loops to H3K9+/- groups look like in biological replicates of H3K9me3 ChIP-seq data?

- Likewise, it is unclear whether high H3K9me3 means overlapping a region enriched in H3K9me3? How far from the CTCF binding site would the H3K9me3 enrichment be? The authors should plot the average profile of H3K9me3 at random regions, regions related to CTCF, enhancers and promoters to show the reader the extent to which the level of this modification coincides with loop formation at CTCF sites in this differentiation system. This question is related to another one – how would the H3K9me3 signal potentially reflect allele specific loop formation?

- Overall, the authors should explain more clearly how the relationship between H3K9me3 would be any more relevant than a possible link with activation of promoters or enhancers at the dynamic loop anchors. (From Figure 1, the inverse relationship between loop formation H3K9me3 and CHD4 enrichment at loop anchors seems the clearest.) Does the loss of H3K9me3 explain more differential loop formation than other statistics? How is differential gene expression compared to H3K9me3? Could a similar conclusion be made have the authors divided the loops based on the signal of H3K27me3 or DNA methylation at anchors?

The link between the dynamics of H3K9me3 and hES cell differentiation is very interesting. Will it hold at later stages of differentiation?

The authors conclude that: “replacement of H3K9me3 by H3K9me2 during pancreatic cell differentiation results in loss of compartmental interactions, suggesting that this simple switch from H3K9me3 to H3K9me2 may lead to the escape of the affected sequences from H3K9me3 biomolecular condensates” without providing evidence that the H3K9me3→H3K9me2 is in fact the cause and not the consequence of any other phenomenon. What if the recruitment of CTCF would outcompete the histone methyltransferase depositing the additional methyl group at K9me2?

The analysis of the pioneer TF is very interesting but purely correlative. Can the authors show ChIP seq for the identified TFs?

The choice of the literature in this rapidly growing field seems brushed over in this paper and this manuscript lacks proper references to papers that have previously shown either the essential phenomena related to cohesin action:

- Schwarzer et al (29094699)
- Wutz (29217591)
- Haarhuis (28475897),

or to loop dynamics in differentiation and thereby already revealing the relationship between loop formation and promoter-enhancer contacts:

- Bonev (29053968)
- Pekowska (30414923)
- Won (27760116)
- Fraser (26700852). For instance, the fact that some loops feature CTCF binding at one or two anchors in the developmental stages preceding loop formation is known.

Line 76 – there the authors should also quote Rao et al., 2014.

It is under investigation how H3K9me3 relates to chromatin compartmentalization and what the mechanism of these interaction would be (whether it is a LLPS for instance). Yet the paper reads as it was already clear. I would stress that in the text.

Reviewer #2 (Remarks to the Author):

This is an interesting and well executed study that has focused on loop activation in a multi-stage differentiation protocol, using various simultaneous orthogonal readouts that provide an in depth characterization of these events. While it is known that minority of CTCF binding and looping is stage/lineage specific, these results show that CTCF loops are unexpectedly dynamic during differentiation, and shed insights into potential mechanisms.

In the end one does wonder what is the relative impact of new CTCF loops in the formation and closure of enhancer domains during differentiation. 2000 new CTCF loops at a single differentiation step could mean that a major fraction of newly formed enhancer clusters are associated with new CTCF loops.

This study is of great interest. I have a few technical comments that need to be addressed in some way, and several optional suggestions

1. This is an extraordinary resource. Can the authors share it in some way other than the raw data? Either with an app, or if that is not possible, with trackhubs or similar highly processed resources. There are many studies that fall beyond the scope of this study and could be explored by others, sometimes starting from the raw data, but also many will want to look up specific loci for which reanalysis is not practical.
2. The intro is very clear and thoughtful, but does not provide an account of what is known about lineage-specific CTCF binding, or new formation of strong loops during differentiation.
3. The differentiation protocol data in the Suppl figures are suitable for this type of study (although the endocrine cells seem to be largely bihormonal). However, it is unclear why qPCRs and immunolocalizations are not provided for all stages. It would be helpful to provide some indication on how well the replicates went.
4. Figure 1A needs to be explained in the legend as it is not a very common representation.
5. In general the changes reported are believable, and match concordant changes in chromatin states and binding patterns. However, the authors could provide some indication of how variable the stage specific changes are in the both replicates. This could give a better sense of how many changes are secondary to technical variability inherent to HiC (e.g. how many new loops are not simply weaker interactions that are more prone to technical variability) or to the SC differentiation protocols.
6. CTCF loops. It would be useful to define at the outset more clearly how they are defined. They do not necessarily have CTCF, in fact many do not bind at one or both anchors. The text states that TADs do not have CTCFs, but it later becomes apparent that these strong loops do not need to be CTCF-bound either. Could another term be used?
7. Based on this data, is there any support for how extrusion could be stopped at non CTCF sites? For example, is there any enrichment of Pol2 or other features at newly formed CTCF- anchors? Even if results are negative this might be mentioned
8. Line 363 makes a general statement about HiChIP enhancer promoter loops, but only supports this with a single example. Can the authors provide quantitative data?
9. The analysis of TFs is interesting, though limited to few datasets. FOXA2 is expressed in most studied stages, but had specificity for new SCbeta loops. This is line with the fact that FOXA2 binds to entirely different sites in different lineages and stages (and is likely to be important at those sites) but needs some explanation.
10. The diagram in Figure 5D was not easy to follow, it would have benefitted from some labeling to indicate what the colors mean (eg change in interactions in X cells?)
11. Discussion. The end has an interesting speculation concerning YY1 enabling cohesin loading to facilitate E interactions with CTCF-bound promoters, but the explanation is only obvious after reading the results section together with this section. This could be written more clearly.

Minor points

The fonts in many figures are impossible to read after a reasonable zoom in a 16 inch screen, see for example 5A labels.

This sentence is not self explanatory as written although later it becomes clear what the authors are saying: "Although both NIPBL and WAPL are enriched at CTCF sites present in loop anchors, loading of cohesin actually happens at different genomic sites via NIPBL-mediated recruitment"

Reviewer #3 (Remarks to the Author):

In this manuscript, Liu and colleagues take advantage of an in vitro pancreatic cell differentiation set up to generate organoids mimicking human pancreatic development. Then, by a combination of epigenomics and Hi-C they focus on the appearance and disappearance of compartmental interactions and CTCF loops during this differentiation process. They provide a potentially useful, high quality, and quite detailed resource for how the 3D genome underlies regulatory changes needed for acquiring the pancreatic cell identity.

Emergence and loss of loops per each stage correlated well with CHD4 loss and gain, respectively, agreeing with previous work on the ChARP complex. Then, compartmental changes are described with addition of more (HiChIP) data.

The most intriguing part of the work is described in Figs 3 and 4, where the nature of differentially-emerging CTCF loops is studied. It appears that new and longer CTCF-anchored loops emerge during differentiation and that often the same anchor can be involved in the formation of different loops (Fig. 3). The explanation of this is, at least in part, that recruitment of stage-specific (oftentimes pioneer) TFs to these sites coincides with loop formation (like FOXA2 - Fig. 4). And this, in turn correlates with de novo gene activation (shown in Fig. 5). These data, though correlative, define a framework that explains gene regulation during pancreatic cell maturation. In my opinion, two things would strengthen this already solid manuscript:

1. A functional experiment showing the necessity of FOXA2 in facilitating CTCF loop establishment (e.g., CRISPR out a FOXA2 motif from under an ATAC peak where a pancreatic islet-specific CTCF peak emerges and follow loop formation with 4C-seq or CaptureC).
2. An analysis of the emergence of longer CTCF loops in later developmental stages in respect to active transcription in/around these loops. This is not to establish gene activation, but to try and interrogate how active transcription might allow longer extrusion patterns given that loop extrusion by cohesin is competing with transcription (based on earlier work from the Corces lab themselves and more recent papers too).

Finally, I left the discussion of the data and analyses behind Fig. 6 as a separate item for three reasons. First, this part of the paper seems detached from the main narrative of the manuscript. Second, it does not offer conclusive evidence for cohesin loading at the respective NIPBL sites (there is some evidence suggesting cohesin-independent function of NIPBL). Third, it leads to the same hypothesis of cohesin loading preferentially at enhancers binding RNAPII as previous papers. Thus, despite the nice analysis, I would suggest moving this figure to the supplement and presenting the data more concisely before discussing those in Fig. 6. Nonetheless, I find the following statement in the Discussion (related to Fig. 6 data): "loading cohesin at active enhancers would ensure that these enhancers are within the same loop as their target promoters" to be insightful.

Secondary remarks:

- Intro: the comment at the end of the first paragraph on "phase-separated membraneless organelles" referring to an explanation of compartmentalisation is still an ambiguous one (based on recent published work from the Rippe lab) and might be best removed or toned-down.
- Fig. 2C: given the signal sparsity in the presented map, I wonder how that top inset qualifies as a loop (I assume called by SIP) compared to the more "canonical" signals of the other two insets? If this

is a 5-kbp resolution map, it might be that using 10-kbp res improves things?
- Results: in many (of not most) places the text is rather general and lacks numbers. Given the resource character of this work, it would be extremely useful for readers to get quantitative information for the many datasets generated and analysed.

A. Papantonis

We thank the reviewers for their thoughtful comments on the manuscript, which have allowed us to prepare an improved revised version. Below we provide a response to each of the comments made by the reviewers. The original comments are displayed in italics and blue font whereas the answers are shown in black regular font. To facilitate reviewing the changes, we provide figures labeled as “R” showing the changes made to the figures in the revised manuscript.

Reviewer #1

“The paper is well written and reveals novel data. The Hi-C profiles of in-vitro differentiating human ES cells to pancreatic cells are a valuable resource in and of its own. But the paper is purely descriptive which somewhat limits its originality (or perhaps should be centered on novel findings and not on a simple description of changes in loop formation). Indeed, it seems that the novel finding would consist in the discovery of a possible link between loss of H3K9me3 and increased CTCF binding which in turn might lead to gain of chromatin loop formation. The authors also suggest that pioneer transcription factors could guide binding of CTCF to new sites in the genome”. We thank the reviewer for the positive comments on the manuscript. We agree that the manuscript describes correlative analyses of genomics data, but the information presented suggests novel and important conclusions for further functional analyses. As the reviewer points out below, others have explored changes in 3D organization in the context of cell differentiation. However, with perhaps the exception of Pekowska et al Cell Systems 2018, these manuscripts have focused on changes in TADs, ~25% of which are not anchored by CTCF and lack a “corner dot”, rather than on loops identified by corner dots and present in a large number of cells in the population. Because of this, previous manuscripts have not explored how CTCF changes during cell differentiation to establish new loops or disassemble old ones to elicit new gene expression patterns. The detailed analysis of possible mechanisms by which CTCF loops form and are eliminated during cell differentiation is a novel contribution of our manuscript. This includes not only de novo occupancy and dismissal of CTCF sites but also differential utilization of existing sites by altering the location of cohesin loading.

*“Overall, the paper would benefit greatly from **more sophisticated computational routines**. It would be extremely beneficial to address differential loop formation using a statistically principled approach. Instead, the authors used a rather enigmatic approach with thresholds without any sound motivation: “Differential loops were identified where at least one sample had a distance normalized Hi-C value ≥ 4 and where there was a least a 4-fold change compared to H9 cells”. How were they further fileted to uncover stage specific loops?”.* We agree with the reviewer that we failed to properly describe the details of how these analyses were performed in the original submission. We have now corrected this oversight and we have added in the Methods section the information on how differential loops were called. A summary of the process is described as follows. For comparisons between different Hi-C libraries, the following normalization steps were taken, 1) valid contacts from each library were randomly picked to match the size of the library with the lowest numbers of contacts; 2) distance normalized by the formula $(\text{observed} - \text{expected}) / (\text{expected} + 1)$; 3) KR normalization was applied to obtain the balanced matrices. The normalized matrices were then used to call differential loops for all stages using the following approach: 1) loops obtained using SIP from all stages were combined; 2) distance and KR normalized contact frequencies in all stages were combined pairwise for all resulting combined loops in step 1; 3) pairwise interaction frequencies from step 2) were used as input in the edgeR R package to identify significant differential loops for each stage (FDR cutoff < 0.1 , p value < 0.05 and fold change ≥ 4). To further identify stage specific loops, the following steps were taken, 1) all differential loops between all stages were combined; 2) contact frequencies were calculated for all stages; 3) loops were ranked by the stages when their contact frequencies reach the maximum (for gained loops) or minimum value (for lost loops); 4) loops were allocated to each stage when they reach maximum changes based on 3) and defined as stage specific loops. To find common loops, combined SIP loops with FDR cutoff ≥ 0.1 , p value ≥ 0.05 or fold change < 4 in edgeR were excluded from stage specific differential loops and defined as common loops.

“There are several analyses that should be included to show more convincingly the relationship between histone modifications, CTCF binding and loop formation – more details below”. This concern is addressed below for each point independently.

“The number 29,905 appears very large provided the sequencing depth, how were the common loops identified and what is the variation in signal strength at these interactions across differentiation timeline? (The word approximately (line 138) should be removed unless the number 29,905 is somewhat imprecise.)”. The total number is indeed large compared to the number of loops found in other publications even those describing datasets with many more valid contacts than ours. A recently published manuscript identified 33,000 loops but the Hi-C dataset included 33 billion contacts (PMID 37280210). We used SIP, instead of HiCCUPS, to call loops with a corner dot. SIP is more efficient than HiCCUPS at calling loops defined by corner dots. Also, although we have less than one billion valid contacts after combining both replicates, the heatmaps have very low background as shown by the several examples shown throughout the manuscript. We have now included the loop calling parameters used to run SIP in the supplemental methods as follows: ‘-norm KR -min 2.0 -max 2.0 -mat 2000 -d 6 -res 5000 -sat 0.01 -t 2500 -nbZero 6 -factor 1 -fdr 0.05 -del true -cpu 48 -factor 4’. We have removed ‘approximately’ from the text as suggested.

“The identification of stage specific loops should be showed more convincingly, the APA profiles should be backed up with histograms showing the distribution of fold changes of loop signal for all loop classes as some profiles of the stage specific loops show substantial Hi-C contacts in the other stages. The authors should also show the changes in loop signal without distance normalization. Such analysis will show beyond doubt that the interactions are genuinely stage specific”. The process for the identification of stage-specific loops is now described in the Methods as mentioned above. We have also included the APA histograms showing the distribution of fold changes of loop signals for all loop classes in Supplementary Figure 3b of the manuscript and in Figure R1 below.

Figure R1

We have also included changes in loop signals without distance normalization in Supplementary Figure 3c and in Figure R2 below. The results are the same as when using distance normalized interactions.

Figure R2

“There are several analyses that should be done to clarify the relationship between chromatin structure and H3K9me3. It is unclear how, in Figure 1, the distinction between loops with high and low level of H3K9me3 was made. The profiles look rather noisy and do not show a clear and uniform gain of H3K9me3. The authors should display $\log_2(\text{fold change})$ of H3K9me3 at loops. How does the plot of H3K9me3 signal against CTCF binding at loop anchors look like? Can one distinguish clear populations of H3K9me3+ and H3K9me3- CTCF anchors? How reproducible are these effects? The authors should show the Hi-C signal for these loops in two replicates of their Hi-C data for each stage separately. What would be the assignment of loops to H3K9+/- groups look like in biological replicates of H3K9me3 ChIP-seq data?” Results described in Figure 1 represent an initial assessment of changes observed at loop anchors. These anchors are 10 kb in size and, in addition to the actual CTCF site that serves as the loop anchor, may also contain enhancers and promoters. An example of the differences in H3K9me3 at loop anchors with or without H3K9me3 in H9 cells is shown below as Figure R3 and it is included in the manuscript as Supplemental Figure 3d.

Figure R3

Figures 4 and 5 describe a more precise analysis of changes taking place at CTCF sites, enhancers, and promoters present at or adjacent to the 10 kb anchors. Therefore, to address the points raised by the reviewer, we have added several panels to the revised Figures 4 and Figure 5 showing \log_2 fold changes in H3K9me3 at stage specific CTCF sites, enhancers, and promoters present at the loop anchors described in Figure 1.

We also show in Figure R4 below, but have not included in the manuscript, differences in Hi-C and H3K9me3 ChIP-seq for each biological replicate.

Figure R4

The reproducibility of the effects is underscored by correlation analyses shown in Figure R5 below for different replicates of each sample corresponding to the different developmental stages. This information has been added to the manuscript in Supp. Figs 2d and 2e.

Figure R5

“Likewise, it is unclear whether high H3K9me3 means overlapping a region enriched in H3K9me3? How far from the CTCF binding site would the H3K9me3 enrichment be? The authors should plot the average profile of H3K9me3 at random regions, regions related to CTCF, enhancers and promoters to show the reader the extent to which the level of this modification coincides with loop formation at CTCF sites in this differentiation system. This question is related to another one – how would the H3K9me3 signal potentially reflect allele specific loop formation?” A plot showing H3K9me3 at CTCF sites, enhancers, promoters and random regions is shown in Figure R6 below and it is included as Supplemental Figure 3e. To further address the concerns of the reviewer, we have included average profiles of H3K9me3 at CTCF sites, enhancers, and promoters in Figures 4 and 5. We have not explored the question of how H3K9me3 signal could potentially reflect allele specific loop formation.

Figure R6

Overall, the authors should explain more clearly how the relationship between H3K9me3 would be any more relevant than a possible link with activation of promoters or enhancers at the dynamic loop anchors. (From Figure 1, the inverse relationship between loop formation H3K9me3 and CHD4 enrichment at lost loop anchors seems the clearest.) Does the loss of H3K9me3 explain more differential loop formation than other statistics? How is differential gene expression compared to H3K9me3? Could a similar conclusion be made have the authors divided the loops based on the signal of H3K27me3 or DNA methylation at anchors? We have clarified the issues brought up by the reviewer in the revised version of the manuscript and performed additional analyses now shown in Figure 4, and Figure 5. As shown in Figure R4 above, there is no strong correlation between the formation of new loop anchors, defined as 10 kb regions, and changes in transcription in the region of the anchors, but changes in transcription can be observed when a more detailed analysis is performed by identifying specific enhancers and promoters within the 10 kb regions. The new panels shown in Figures 4 and 5 indicate that changes in H3K27me3 and H3K9me3 similarly correlate with recruitment of CTCF and enhancer activation but not with promoter activation. However, the correlation and magnitude of changes are stronger for H3K9me3. We have also divided the CTCF anchors into those overlapping peaks of H3K9me3 called by MACS2 and those overlapping H3K27me3 peaks. CTCF loop anchors overlapping each histone modification also contain smaller amounts of the second one but the overall pattern of changes during pancreatic cell differentiation is similar to that found for the combined CTCF anchors. Changes in DNA methylation correlate better with those in H3K9me3 than in H3K27me3. This information is shown below in Figure R7 and in Supplementary Figs. 5a and 5b.

Figure R7

The link between the dynamics of H3K9me3 and hES cell differentiation is very interesting. Will it hold at later stages of differentiation? From the results shown in Figures 1c and 1e, we observe that changes of H3K9me3 at loop anchors are stage specific. At any given stage, active loop anchors lose H3K9me3 but they gain it again in the following stage when the anchors become inactive at later stages. This is also shown now in more detail in Figure 4 by focusing on specific CTCF sites, enhancers, and TSSs present in the 10 kb loop anchors.

The authors conclude that: “replacement of H3K9me3 by H3K9me2 during pancreatic cell differentiation results in loss of compartmental interactions, suggesting that this simple switch from H3K9me3 to H3K9me2 may lead to the escape of the affected sequences from H3K9me3 biomolecular condensates” without providing evidence that the H3K9me3→H3K9me2 is in fact the cause and not the consequence of any other phenomenon. What if the recruitment of CTCF would outcompete the histone methyltransferase depositing the additional methyl group at K9me2? We agree with the reviewer that we are currently unable to make causal inferences from our data. We have rephrased the statement as: “replacement of H3K9me3 by H3K9me2 during pancreatic cell differentiation correlates with loss of compartmental interactions. It is thus possible that the simple switch from H3K9me3 to H3K9me2 may lead to the escape of the affected sequences from H3K9me3 biomolecular condensates”.

The analysis of the pioneer TF is very interesting but purely correlative. Can the authors show ChIP seq for the identified TFs? We have only performed ChIP-seq with FOXA2 because, based on the literature, this is the pioneer factor most likely to play a role during the differentiation of endodermal lineages. We now include analyses of FOXA2 changes at CTCF sites for all stages in Figure 4.

The choice of the literature in this rapidly growing field seems brushed over in this paper and this manuscript lacks proper references to papers that have previously shown either the essential phenomena related to cohesin action:
 - Schwarzer et al (29094699)

- Wutz (29217591)
- Haarhuis (28475897),
or to loop dynamics in differentiation and thereby already revealing the relationship between loop formation and promoter-enhancer contacts:
- Bonev (29053968)
- Pekowska (30414923)
- Won (27760116)
- Fraser (26700852). For instance, the fact that some loops feature CTCF binding at one or two anchors in the developmental stages preceding loop formation is known.
Line 76 – there the authors should also quote Rao et al., 2014.
It is under investigation how H3K9me3 relates to chromatin compartmentalization and what the mechanism of these interaction would be (whether it is a LLPS for instance). Yet the paper reads as it was already clear. I would stress that in the text. As the reviewer suggested, we have added the above references in the revised version. We also stressed that the relationship between H3K9me3 and chromatin compartmentalization and the underlying mechanisms are not yet understood.

Reviewer #2

This is an interesting and well executed study that has focused on loop activation in a multi-stage differentiation protocol, using various simultaneous orthogonal readouts that provide an in depth characterization of these events. While it is known that minority of CTCF binding and looping is stage/lineage specific, these results show that CTCF loops are unexpectedly dynamic during differentiation, and shed insights into potential mechanisms.

In the end one does wonder what is the relative impact of new CTCF loops in the formation and closure of enhancer domains during differentiation. 2000 new CTCF loops at a single differentiation step could mean that a major fraction of newly formed enhancer clusters are associated with new CTCF loops.

This study is of great interest. I have a few technical comments that need to be addressed in some way, and several optional suggestions.

We thank the reviewer for the positive comments on the manuscript.

1. *This is an extraordinary resource. Can the authors share it in some way other than the raw data? Either with an app, or if that is not possible, with trackhubs or similar highly processed resources. There are many studies that fall beyond the scope of this study and could be explored by others, sometimes starting from the raw data, but also many will want to look up specific loci for which reanalysis is not practical.* As the reviewer suggested, we have uploaded most datasets in this study to UCSC link https://genome.ucsc.edu/cgi-bin/hgTracks?db=hg38&lastVirtModeType=default&lastVirtModeExtraState=&virtModeType=default&virtMode=0&nonVirtPosition=&position=chr2%3A25160915%2D25168903&hgsid=1674604212_cMBhoSEw36ot7IAhBJD1DUuNgrLL and included a copy of the link in the revised manuscript in that Data Availability section.

2. *The intro is very clear and thoughtful, but does not provide an account of what is known about lineage-specific CTCF binding, or new formation of strong loops during differentiation.* We have now included in the Introduction several sentences describing what is known about lineage-specific CTCF binding. Most manuscript describing changes in 3D organization during differentiation have focused the analyses on changes in TADs defined based on changes in the directionality of interactions rather than changes in loops defined by corner dots.

3. *The differentiation protocol data in the Suppl figures are suitable for this type of study (although the endocrine cells seem to be largely bihormonal). However, it is unclear why qPCRs and immunolocalizations are not provided for all stages. It would be helpful to provide some indication on how well the replicates went.* To compare the expression levels of stage specific marker genes throughout all differentiation stages, we have analyzed the transcription of typical marker genes throughout all stages

from RNA-seq data. The results are now shown in Supplementary Figure 2b and also in Figure R8 below.

Figure R8

To determine the reproducibility between replicates, we plotted the correlation of TPM values of genes from RNA-seq data of biological replicates in Supplementary Figures 2a and 2c as shown in Figure R9 below.

Figure R9

4. Figure 1A needs to be explained in the legend as it is not a very common representation. Response: We have now explained this panel in more detail in the figure legend.

5. In general the changes reported are believable and match concordant changes in chromatin states and binding patterns. However, the authors could provide some indication of how variable the stage specific changes are in the both replicates. This could give a better sense of how many changes are

secondary to technical variability inherent to HiC (e.g. how many new loops are not simply weaker interactions that are more prone to technical variability) or to the SC differentiation protocols. Response: As the reviewer suggested, we have performed a similar analysis to that shown in Figures 1g and 1h using each replicate separately and the results are shown in Figure R4 above. The results suggest that the stage specific changes are consistent in both replicates and not secondary to technical variability inherent to Hi-C or to the pancreatic cell differentiation protocols. Correlation of Hi-C data between replicates are shown in Figure R5 above and in Supplementary Figs. 2d and 2e.

6. CTCF loops. It would be useful to define at the outset more clearly how they are defined. They do not necessarily have CTCF, in fact many do not bind at one or both anchors. The text states that TADs do not have CTCFs, but it later becomes apparent that these strong loops do not need to be CTCF-bound either. Could another term be used? Response: We agree with the reviewer that this is a problem, and we are open to suggestions on how to deal with this issue. To avoid adding more terms to this already confusing field, we have maintained the name “CTCF loops” but most times we use the term “loop”. We have made very clear in multiple places in the manuscript that the anchors do not always have CTCF. If the reviewer does not agree with this decision, we are equally open to only using the term “contact loops”, but we are concerned that most readers would be confused and wonder if they are the same as CTCF loops or not.

7. Based on these data, is there any support for how extrusion could be stopped at non CTCF sites? For example, is there any enrichment of Pol2 or other features at newly formed CTCF- anchors? Even if results are negative this might be mentioned. Response: As the reviewer suggests, we have included the signals of Pol2 ChIP-seq and ATAC-seq on non-CTCF new loop anchors in Supplementary Figure 6b and in Figure R10 below. Assuming that cohesin extrusion at these 10 kb anchors is stopped by proteins bound to DNA, we combined all ATAC-seq peaks present at non-CTCF anchors and we identified motifs of TFs enriched at the summits of these ATAC-seq peaks. The results suggest that various TF motifs are enriched at these putative anchors, including pioneer and standard TFs (Supplemental Figure 6a and Figure R9 below). Both SOX2 and RNAPII are also enriched (Supplementary Figure 5b and Figure R10). However, the interpretation of these results is not straightforward, and we have refrained from making strong claims in the manuscript. Because of the limitations in the resolution of our Hi-C data, anchors cannot be defined as less than 10 kb long. When a CTCF site is located within the 10 kb, we assume that this site is responsible for stopping cohesin extrusion and forming the loop. When CTCF is absent, the anchors may contain different TFs bound to sequences in enhancers and promoters, precluding the identification of the precise proteins stopping cohesin extrusion. It is possible that specific TFs are involved in this process. Based on recent results, it is more likely that transcribing RNAPII in combination with other large protein complexes interferes with cohesin extrusion and they can act as loop anchors. This possibility is now discussed in the manuscript.

Figure R10

8. Line 363 makes a general statement about HiChIP enhancer promoter loops, but only supports this with a single example. Can the authors provide quantitative data? We apologize for not properly explaining Figures 5d and 5e, which this comment refers to. This figure represents a meta-analysis of all enhancer-promoter interactions contained within CTCF loops. Figure 5d represents 158 CTCF loops and 195 enhancer-promoter interactions. Figure 5e includes 426 enhancer-promoter interactions contained within 321 CTCF loops. We have now added more information in the figure legend to explain the contents of the figure.

9. The analysis of TFs is interesting, though limited to a few datasets. FOXA2 is expressed in most studied stages, but had specificity for new SCbeta loops. This is in line with the fact that FOXA2 binds to entirely different sites in different lineages and stages (and is likely to be important at those sites) but needs some explanation. Response: We agree with the reviewer that additional work will be required to understand the preference of FOXA2 for new loops in SC-β cells. It is possible that other pioneer factors are preferentially used at other stages. Answering these questions will require examining changes in the distribution of the different factors identified to be enriched at new loop anchors. We expect to perform these experiments in the future, and we hope that the reviewer will agree that these experiments are outside the scope of this manuscript.

10. The diagram in Figure 5D was not easy to follow, it would have benefitted from some labeling to indicate what the colors mean (eg change in interactions in X cells?). Response: We have added arrowheads and described the contents of this figure in the legend to facilitate understanding of the results as the reviewer suggested.

11. Discussion. The end has an interesting speculation concerning YY1 enabling cohesin loading to facilitate E interactions with CTCF-bound promoters, but the explanation is only obvious after reading the results section together with this section. This could be written more clearly. Response: We have rewritten both parts more clearly without adding too much more information, since these statements are somewhat speculative.

Minor points

The fonts in many figures are impossible to read after a reasonable zoom in a 16 inch screen, see for example 5A labels. Response: We have increased the size of the fonts in all panels of Figure 5 and Supplemental Figure 7, which were the ones affected by this problem.

This sentence is not self explanatory as written although later it becomes clear what the authors are saying: "Although both NIPBL and WAPL are enriched at CTCF sites present in loop anchors, loading of cohesin actually happens at different genomic sites via NIPBL-mediated recruitment". Response: We have modified this sentence, which now read as follows "Loading of the cohesin complex takes place at genomic sites different from loop anchors via NIPBL-mediated recruitment".

Reviewer #3

1. A functional experiment showing the necessity of FOXA2 in facilitating CTCF loop establishment (e.g., CRISPR out a FOXA2 motif from under an ATAC peak where a pancreatic islet-specific CTCF peak emerges and follow loop formation with 4C-seq or CaptureC). Response: We agree with the reviewer that the inclusion of functional experiments such as the one suggested would allow us to make causal, instead of correlative, inferences. Dr. Xiaowen Liu, the first author of the manuscript, has recently obtained an independent faculty position and she is in the process of carrying out this line of work in her own lab. As a consequence, the lab of the corresponding author is not currently working on this topic. We hope that the reviewer will agree that these additional experiments will take too much time to complete to be able to include them in the manuscript.

2. An analysis of the emergence of longer CTCF loops in later developmental stages in respect to active transcription in/around these loops. This is not to establish gene activation, but to try and interrogate how active transcription might allow longer extrusion patterns given that loop extrusion by cohesin is competing with transcription (based on earlier work from the Corces lab themselves and more recent papers too). This is a very interesting suggestion and we have performed the analyses suggested by the reviewer. Unfortunately, the results are inconclusive, and we have not included them in the manuscript. Figure R11 below summarizes the observations. In 12-17% of new loops formed by an existing anchor (dark blue), the genes between the old and new anchors are transcribed in both directions but genes transcribed towards the new anchor are upregulated in the new stage. In 4-8% of cases (orange color), all genes between the old and new anchors are transcribed towards the new anchor and they are upregulated in the new stage. In 4-11% of cases (grey color), all genes between the old and new anchors are transcribed towards their old anchor but their expression is downregulated in the new stage. These three cases agree with the hypothesis that transcription plays a role in the formation of extended loops, but they only represent 25-30% of the total cases. In 31% of cases (yellow color), there are no genes located between the old and new anchors, and in 39-44% of cases (Other, light blue), the arrangement of genes and their expression is the opposite to what the hypothesis predicts i.e., genes upregulated pointing towards old anchors, genes downregulated pointing towards the new anchors, non-transcribed genes, etc. Therefore, although transcription may play a role in the formation of a subset of new loop anchors, there may also be other mechanisms at play. Because of this, we have left these results out of the revised version of the manuscript but we agree that this idea is worth exploring in more detail in the future

Figure R1

Finally, I left the discussion of the data and analyses behind Fig. 6 as a separate item for three reasons. First, this part of the paper seems detached from the main narrative of the manuscript. Second, it does not offer conclusive evidence for cohesin loading at the respective NIPBL sites (there is some evidence suggesting cohesin-independent function of NIPBL). Third, it leads to the same hypothesis of cohesin loading preferentially at enhancers binding RNAPII as previous papers. Thus, despite the nice analysis, I would suggest moving this figure to the supplement and presenting the data more concisely before discussing those in Fig. 6. Nonetheless, I find the following statement in the Discussion (related to Fig. 6 data): "loading cohesin at active enhancers would ensure that these enhancers are within the same loop as their target promoters" to be insightful. Response: We agree with the reviewer on these points i.e., it is not completely clear that NIPBL is the loader for cohesin, cohesin may load without NIPBL, and NIPBL is already known to be present at enhancers/promoters. However, we do think the results in Figure 6 follow naturally from the rest of the manuscript because they represent one possible explanation for the formation of extended loops in the transition between differentiation states. As the reviewer mentioned in previous comments, it is possible that preference for specific anchors is determined by transcription of genes in the vicinity of the CTCF. For example, transcription away from the previous anchor and towards the new anchor responsible for the formation of extended loops could explain the formation of these new larger loops. However, we have been unable to find correlations between the formation of extended loops and the direction of transcription of adjacent sequences to support this hypothesis (see answer to previous comment). An alternative explanation is the use of different sites for cohesin loading and current Figure 6 shows results supporting this concept. We think this is an interesting idea that has not been given sufficient attention by the field and would explain why there are changes in loops without changes in CTCF distribution. Additional work will have to be done to properly test this idea and we hope the publication of the information in Figure 6 will encourage other labs to study this issue in detail.

Secondary remarks:

Intro: the comment at the end of the first paragraph on "phase-separated membraneless organelles" referring to an explanation of compartmentalisation is still an ambiguous one (based on recent published work from the Rippe lab) and might be best removed or toned-down. Response: We have deleted the words "phase-separated" from this sentence.

Fig. 2C: given the signal sparsity in the presented map, I wonder how that top inset qualifies as a loop (I assume called by SIP) compared to the more "canonical" signals of the other two insets? If this is a 5-kbp resolution map, it might be that using 10-kbp res improves things? Response: The old map in Fig. 2C is at 5 kb resolution. A map using 10 kb resolution is shown in Figure R11 below. This new heatmap does not seem to improve the signal and we have left the original map in the figure as it was. We can replace it with this one if the reviewer feels this would be an improvement.

Figure R12

Results: in many (if not most) places the text is rather general and lacks numbers. Given the resource character of this work, it would be extremely useful for readers to get quantitative information for the many datasets generated and analysed. Response: We have now added more quantitative information throughout the text in the revised version. Some of this information is also present in the new supplementary figures.

REVIEWERS' COMMENTS

Reviewer #1 (Remarks to the Author):

I would like to thank the authors for so nicely addressing the questions I raised. The additional analyses strengthen the message and clarify it at places that were somewhat unclear.

One thing remains in my opinion uncommon. Why did the authors perform matrix balancing (KR) on the matrices that were distance normalized (Lines: 794-795 also in the rebuttal letter)? It misses the point; the KR should be performed before the distance normalization.

Some additional remarks:

- The M&M section Analysis of ChIP-seq and CUT&Tag data requires further polishing "PCR duplicates were removed manually" should be rephrased.
- It would be great to spell out in the text how many H3K9me3+ and - loops there were, I would also add text in Figure 1c indicating H3K9me3- and H3K9me3+ loops
- Figure 4B – I am confused about what is displayed in this panel.
- Figure 4c – the tick labels and x axis label are overlapping.

Reviewer #2 (Remarks to the Author):

The authors have done a very exhaustive revision that addresses all my comments. I clicked on the link but it does not work for me. I encourage the authors to do a UCSC trackhub, which will not cost much effort but will be of interest to beta cell and gene regulation scientists.

Reviewer #3 (Remarks to the Author):

I wish to thank the authors for the detailed responses to all comments (incl. mine). I find that their arguments and additional analyses address most concerns, and I also do respect their response about functional work on the FOXA transcription factor being pursued by a newly-established lab. I would like to see the manuscript published in its current form.

A. Papantonis

Response to reviewer's comments

We have incorporated the comments of the reviewers in the new revised version as follows:

Reviewer #1

1. One thing remains in my opinion uncommon. Why did the authors perform matrix balancing (KR) on the matrices that were distance normalized (Lines: 794-795 also in the rebuttal letter)? It misses the point; the KR should be performed before the distance normalization. Thank you for bringing this up again. We did perform KR normalization first followed by distance normalization. I made a mistake when re-writing a very long description in an earlier draft of the manuscript describing this in the Methods and then I copied pasted from the methods into the response document. This mistake has now been corrected in the Methods section as follows: "For comparisons between different Hi-C libraries, the following normalization steps were taken, 1) valid contacts from each library were randomly picked to match the size of the library with the lowest numbers of contacts; 2) KR normalization was applied to obtain the balanced matrices. 3) Matrices were then distance normalized by the formula $(\text{observed} - \text{expected}) / (\text{expected} + 1)$. The normalized matrices were then used to call differential loops of all stages".

Some additional remarks:

- The M&M section Analysis of ChIP-seq and CUT&Tag data requires further polishing "PCR duplicates were removed manually" should be rephrased. We have rephrased the sentence in the Methods sections to "PCR duplicates were removed using Picard Tools (<http://picard.sourceforge.net;https://broadinstitute.github.io/picard/>)".

- It would be great to spell out in the text how many H3K9me3+ and – loops there were, I would also add text in Figure 1c indicating H3K9me3- and H3K9me3+ loops. We have now included a sentence in the results and in the legend to Fig 1C stating that "These results show that 250 stage-specific CTCF loops contain H3K9me3 whereas 5033 lack this histone modification".

- Figure 4B – I am confused about what is displayed in this panel. Figure 4B shows the number of binding sites for different transcription factors present in 10 kb regions containing new CTCF loop anchors that increase in interaction frequency at different stages of pancreatic cell differentiation. We have re-written the legend for Fig. 4b to now read "Frequency of binding motifs for various transcription factors found at the summits of ATAC-TF peaks present within 10 kb regions containing anchors of CTCF loops showing increased interactions at different stages of pancreatic cell differentiation".

- Figure 4c – the tick labels and x axis label are overlapping. We have moved down the X axis label in Fig 4c to avoid the overlap with the tick labels.

Reviewer #2

The authors have done a very exhaustive revision that addresses all my comments. I clicked on the link but it does not work for me. I encourage the authors to do a UCSC trackhub, which will not cost much effort but will be of interest to beta cell and gene regulation scientists. We apologize for the problem. We now include a link that is working. The new link is <https://genome.ucsc.edu/cgi-bin/hgTracks?db=hg38&lastVirtModeType=default&lastVirtModeExtraState=&virtModeType=def>

http://www.ncbi.nlm.nih.gov/nuccore/1711140546?term=4oiO7H0pztEcMSrKRlpnSCYhbDET&from_uid=1711140546&virtMode=0&nonVirtPosition=&position=chr2%3A25160915%2D25168903&hgsid=1711140546_4oiO7H0pztEcMSrKRlpnSCYhbDET and it has been included in the manuscript in the Data Availability section.